# Sensitivity of tropospheric ozone to halogen chemistry in the chemistry-climate model LMDZ-INCA vNMHC

Cyril Caram[1], Sophie Szopa[1], Anne Cozic[1], Slimane Bekki[2], Carlos A. Cuevas[3] and Alfonso Saiz-Lopez[3]

[1]Laboratoire des Sciences du Climat et de l'Environnement, LSCE/IPSL, CEA-CNRS-UVSQ, Université Paris-Saclay, Gif-sur-Yvette, France
[2]Laboratoire Atmosphère, Milieux, Observations Spatiales, Institut Pierre Simon Laplace, LATMOS/IPSL, CNRS-UVSQ-Sorbonne Université, Guyancourt and Paris, France
[3]Department of Atmospheric Chemistry and Climate, Institute of Physical Chemistry Rocasolano, CSIC, Madrid, Spain

*Correspondence to*: Sophie Szopa (sophie.szopa@lsce.ipsl.fr)

**Abstract.** The atmospheric chemistry of halogenated species (Cl, Br, I) participates in the global chemical sink of tropospheric ozone and perturbs the oxidizing capacity of the troposphere, notably by influencing the atmospheric lifetime of methane. Global chemistry-climate models are commonly used to assess the global budget of ozone, its sensitivity to emissions of its precursors, and to project its long-term evolution. Here, we report on the implementation of tropospheric sources and chemistry of halogens in the chemistry-climate model LMDZ-INCA and evaluate halogen effects on the tropospheric ozone budget. Overall, the results show that the model simulates satisfactorily the impact of halogens on the photooxidizing system in the troposphere, in particular in the marine boundary layer. To quantify the effects of halogen chemistry in LMDZ-INCA, standard metrics representative of the behavior of the tropospheric chemical system ($O_x$, $HO_x$, $NO_x$, $CH_4$, and NMVOCs) are computed with and without halogens. Addition of tropospheric halogens in the LMDZ-INCA model leads to a decrease of 22% in the ozone burden, 8% in OH, and 33% in NOx. Sensitivity simulations show for the first time that the inclusion of halogen chemistry makes ozone more sensitive to perturbations in $CH_4$, $NO_x$, and NMVOCs. Consistent with other global model studies, the sensitivity of the tropospheric ozone burden to changes from pre-industrial to present-day emissions is found to be ~20% lower when tropospheric halogens are taken into account.

## 1 Introduction

The atmospheric chemistry of halogenated species was first studied for its importance for the composition of the stratosphere and in particular the ozone layer (Molina and Rowland, 1974; Stolarski and Cicerone, 1974), but its importance for tropospheric chemistry was also suggested as soon as the early 1980s (Chameides and Davis, 1980). With the progress of measurement techniques, marine aerosols were thought to harbor and release reactive halogens into the gas-phase (Cicerone, 1981). Later on, halogenated species were found to be responsible for ozone destruction in the Arctic troposphere (Bottenheim et al., 1990; Oltmans and Komhyr, 1986). Meanwhile, natural and anthropogenic sources of halogens were starting to be identified and characterized (Graedel and Keene, 1995; Lobert et al., 1999; McCulloch et al., 1999; Carpenter, 2003; Platt and Hönninger, 2003; Saiz-Lopez and Plane, 2004). In recent years, the addition of halogenated chemistry into global chemistry-transport models have revealed their significant influence on the photo-oxidizing chemistry of the troposphere on a global scale (Saiz-Lopez et al., 2012a; Simpson et al., 2015). This influence arises from the high reactivity of atomic halogens (e.g., Cl, Br, I) and halogen oxide radicals (e.g., ClO, BrO, IO and higher oxides), which are produced in the atmosphere by the decomposition of very short-lived halogenated substances (VSL), by heterogeneous reactions on sea salt aerosols, and, in the case of iodine, also released via ozone deposition to the ocean surface. They can also be generated by polar specific halogen activation mechanisms on sea ice surfaces, blowing snow and the snowpack (as reviewed by Abbatt et al. 2012).

Details on the tropospheric chemistry of chlorine, bromine and iodine can be found in review articles (e.g. Saiz-Lopez and von
Glasow, 2012; Simpson et al., 2015). Reactive halogen species mainly affect tropospheric chemistry by (1) modifying the partitioning of $HO_x$ ($HO_2$ + OH) and nitrogen oxides ($NO_x$ = $NO_2$ + NO) and reacting with $NO_2$ to form species that remove NOx from the system via deposition or other loss processes; (2) being involved in the oxidation of non-methane volatile organic compounds (NMVOCs) and mercury; (3) participating in the formation of new particles (Giorgi and Chameides, 1985; O'Dowd et al., 2002; von Glasow et al., 2004; Gómez Martín et al., 2020; Saiz-lopez et al., 2012); and (4) constituting a sink
for tropospheric ozone at the global scale. Halogenated species play a role in the ozone and hydroxyl radical (OH) budgets and therefore influence the oxidative capacity of the troposphere (Iglesias-Suarez et al., 2020, Badia et al. 2021). In addition, by influencing the abundances of tropospheric ozone, which is an important greenhouse gas, halogens can indirectly affect the climate (Saiz-Lopez et al., 2012b; Hossaini et al., 2015; Sherwen et al., 2017).

The loss of tropospheric ozone by halogens is mainly induced by iodine and bromine species with a lower contribution of
chlorine species (Sherwen et al., 2016a). It can be caused by direct catalytic ozone destruction and reduction in ozone production through the removal of $NO_2$ by halogen radicals. Halogen chemistry is thought to be especially active in pristine atmospheres that host 60 to 70% of the global burden of tropospheric ozone (Holmes et al., 2013). Based on model calculations, the halogen-driven ozone loss is estimated to be around 30% in the marine boundary layer (von Glasow et al., 2002, 2004; Saiz-Lopez et al., 2014; Sherwen et al., 2016a) and 5% to 20% globally (Yang et al., 2005; Saiz-Lopez et al., 2012b, 2014;
Sherwen et al., 2016a), going up to 25 to 30% in Badia et al., 2019. Consequently, it might be important to consider halogens in chemistry-climate modelling. It could also help to reconstruct more reliably past atmospheric compositions (Saiz-Lopez et al., 2014; Young et al., 2013).

Despite its potential importance, only three global chemistry models have so far implemented detailed representations of halogenated (Cl, Br and I) tropospheric chemistry: CAM-Chem (Saiz-Lopez et al. 2012), GEOS-Chem (Sherwen 2016a) and
TOMCAT (Hossaini et al. 2015). Halogen chemistry is complex with a high computational cost (Fernandez et al. 2020), Earth System Models used to assess how the evolution of atmospheric composition influences climate need to find the right compromise between the level of details (e.g. number of species and processes) here for halogen chemistry representations and the computing efficiency (Box 6.1 in Szopa et al. 2021). Indeed, the computational cost has to be sufficiently low for multiple long chemistry-climate simulations. So far, none of the chemistry-climate models involved in recent large
intercomparison projects included tropospheric chemistry of very short-lived halogens and hence the impacts of their chemistry on key short-lived climate forcers (such as ozone and methane). Such multi-model projects are important. They are used to assess the budget of key chemical species, which can impact directly or indirectly climate or global air quality, and their sensitivity to changes in anthropogenic and natural emissions. The results are also used to evaluate the effects of international policies (e.g. Protocol to Abate Acidification, Eutrophication and Ground-level Ozone called Gothenburg Protocol) by
quantifying source-receptor relations at continental scales (in the case of Hemispheric Transport of Air Pollution project, HTAP, Fiore et al., 2009), to explore the sensitivity of short lived climate forcers to changes in emissions of their precursors since the preindustrial period, or to project their possible future evolutions under a range of scenario assumptions (in the case of Atmospheric Chemistry and Climate Model Intercomparison Project (ACCMIP, Young et al., 2013) or, more recently, the Aerosol Chemistry Model Intercomparison Project (AerChemMIP, Stevenson et al., 2020).

The primary purpose of this work is to assess whether iodine, bromine and chlorine chemistry can, beyond their already established effect on global tropospheric ozone budget, affect the sensitivity of ozone to the changes of its precursors. In order to do so, we implemented a simplified representation of tropospheric halogen chemistry in a global chemistry-climate model, the LMDZ-INCA model (Laboratoire de Météorologie Dynamique (LMD) General Circulation Model, LMDZ; Interactions with Chemistry and Aerosols, INCA) with a focus on reproducing the overall effect of halogen chemistry on global
tropospheric ozone. Results of the 3 global tropospheric chemistry models that already include detailed representations of this chemistry are used to guide the choices of the chemical reactions and emissions in our simple halogen scheme, and to assess

the LMDZ-INCA simulations. First, we evaluate the ability of the model to simulate the global tropospheric budget of halogens and their effect on ozone by using the results of the CAM-Chem and GEOS-Chem models as references. The limitations of our model are also discussed, notably how some simplifications in our model affect the simulation of the spatial distribution of some specific halogenated species. Second, we investigate how the sensitivities of ozone, methane and hydroxyl radical to perturbations in anthropogenic emissions of photooxidant precursors change when halogen chemistry is accounted for. For instance, we explore changes in ozone burden and net destruction in simulations considering present-day (PD) and pre-industrial (PI) emissions with and without halogen chemistry. Finally, the last section recalls key findings and we conclude with implications for chemistry-climate simulations of tropospheric ozone evolution.

## 2 Methodology

### 2.1 Model description

#### 2.1.1 LMDZ-INCA

LMDZ is an atmospheric general circulation model developed by Sadourny and Laval (1984). The primitive equations of the GCM are solved with a time step of 3 minutes, a large-scale transport of tracers is carried out every 15 minutes and the physical processes are calculated at a time interval of 30 minutes. The LMDZ model provides at each time step the physical quantities (temperature, pressure, water vapor, turbulence, boundary layer, etc.) needed to compute the transport and chemistry in INCA.

INCA is an atmospheric chemistry model developed at LSCE (Laboratoire des Sciences du Climat et de l'Environnement). The model takes into account the primary emissions from natural sources or by anthropogenic activities. It computes physical processes such as dry and wet deposition and photochemical reactions (Hauglustaine et al., 2004). The atmospheric concentration fields are integrated in time by solving differential equations when called by LMDZ, with a time step of 30 minutes. For this development, we used a version of INCA (version v_5.2.1) simulating interactively the gaseous photochemistry whereas aerosols distribution obtained with a more complex version of INCA are prescribed through climatologies. This version represents in particular the chemistry of the inorganic species $O_x$, $NO_x$ and $HO_x$ as well as $CH_4$, CO, non-methane volatile organic compounds (NMVOCs) and their photooxidized products (Folberth et al., 2006).

Biogenic emissions like those of isoprene are provided by the global vegetation model, ORCHIDEE (Organising Carbon and Hydrology In Dynamic Ecosystems) (Folberth et al., 2006). Anthropogenic emissions are based on the CEDS v2016-07-2 emission inventory for the year 2010 (Hoesly et al., 2018).

The LMDZ-INCA model has been used and assessed in numerous previous model intercomparison exercises aimed at simulating the preindustrial, present and future tropospheric composition (e.g. Szopa et al., 2013), the long range transport of air pollution in the HTAP exercice (Fiore et al., 2009) or the chemistry/climate interactions in ACCMIP (Naik et al., 2013).

#### 2.1.2 Halogen chemistry

The chemistry of halogenated compounds implemented in LMDZ-INCA is essentially based on the scheme developed for CAM-Chem (Fernandez et al., 2014b; Ordóñez et al., 2012; Prados-Roman et al., 2015a; Saiz-Lopez et al., 2012b, 2014, 2015) and GEOS-Chem (Sherwen et al. 2016b), with a priority given to the reactions of importance for the ozone sinks (Figure 1).

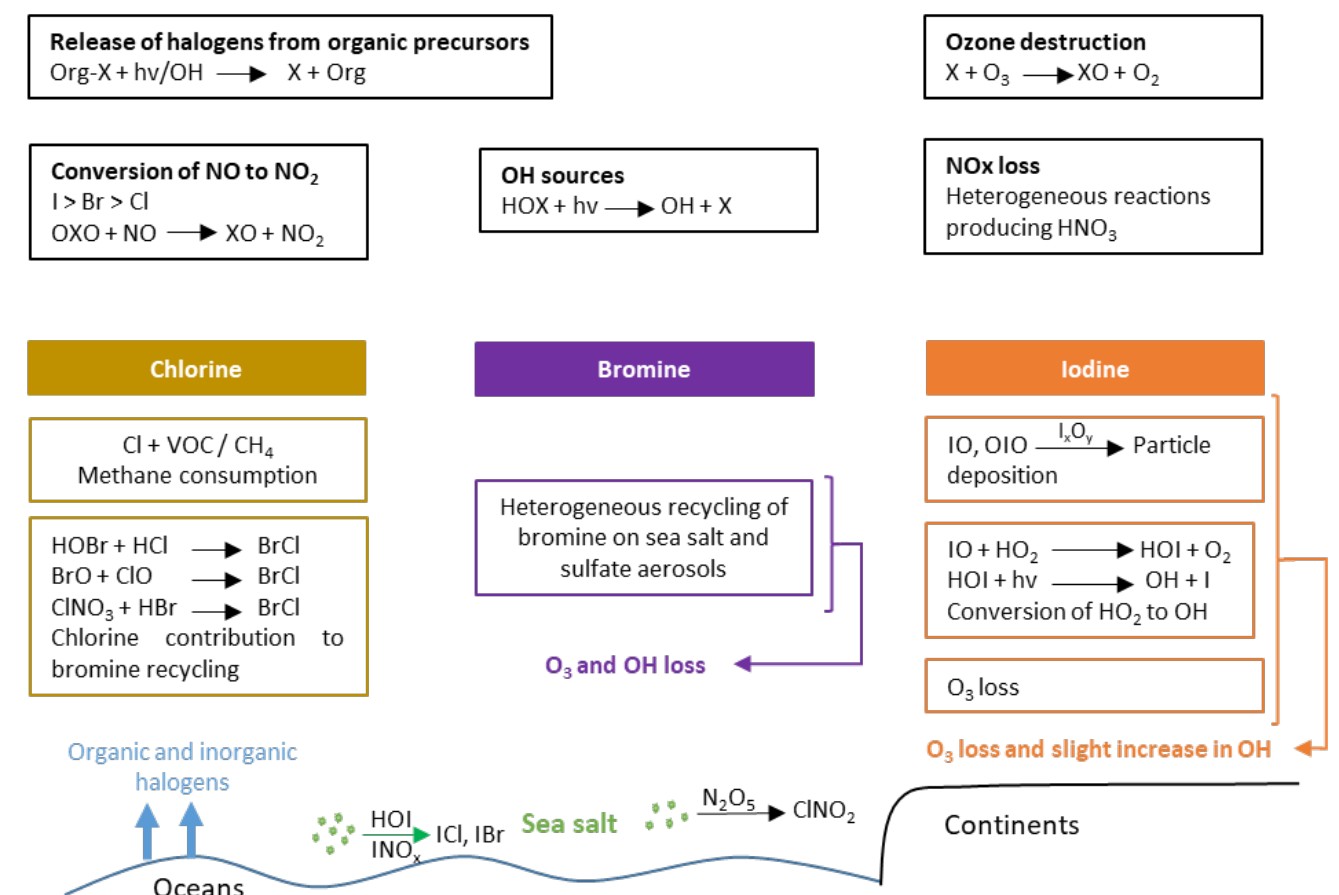


**Figure 1: Schematic of the most important tropospheric chemical processes involving halogenated compounds (containing Cl, Br, I) as represented in LMDZ-INCA.**

A total of 42 halogenated species have been added to the INCA chemical scheme (Table 1). The list of the 160 chemical reactions added to the INCA chemical scheme can be found in Tables 2, 3, 4 and 5. Photolysis reactions of halogens as well as relative quantum yields and absorption cross-sections are listed in Table 2. Uptake coefficients for heterogeneous reactions are based on experimental values and data selected for other models and are reported in Table 5. Recycling through heterogeneous chemistry on sea salt and sulfate aerosols is included while the heterogeneous bromine reactions in water droplets and on ice crystals in clouds are not. Dry deposition is calculated with the Wesely scheme (Wesely, 1989) while washout of gases by precipitation is simulated by using Henry's law constants detailed in Table S1 (Supplementary Material). Some VSL halocarbons, $CH_2BrCl$, $CHBr_2Cl$ and $CHBrCl_2$, represent a few percent of the emissions of halocarbons (Ordóñez et al. 2012). Considering the high uncertainty in their emissions and their limited abundances, we chose not to consider their chemistry in this version.


**Table 1. List of halogen tracers in INCA.**

| Tracer | Name | Processes affecting the tracer | | | | |
|--------|------|--------------------------------|--|--|--|--|
| | | chemistry | advection/convection | emissions [a] | Dry deposition | Wet deposition |
| **Br₂** | Dibromine | ● | ● | - | ● | ● |
| **Br** | Bromine | ● | - | - | - | - |
| **BrO** | Bromine monoxide | ● | - | - | - | - |
| **HOBr** | Hypobromous acid | ● | ● | - | ● | ● |
| **HBr** | Hydrogen bromide | ● | ● | - | ● | ● |
| **BrNO₂** | Bromine nitrite | ● | ● | - | ● | ● |

| Formula | Name | | | | | |
|---|---|---|---|---|---|---|
| **BrNO₃** | Bromine nitrate | ● | ● | - | ● | ● |
| **CH₃Br** | Bromomethane | ● | ● | ● | - | - |
| **CH₂Br₂** | Dibromomethane | ● | ● | ● | - | - |
| **CHBr₃** | | ● | ● | ● | ● | |
| **I₂** | Diiodine | ● | ● | ● | ● | ● |
| **I** | Iodine | ● | - | - | - | - |
| **IO** | Iodine monoxide | ● | - | - | - | - |
| **OIO** | Dioxidoiodine | ● | - | - | - | - |
| **HOI** | Hypoiodous acid | ● | ● | ● | ● | ● |
| **HI** | Hydrogen iodide | ● | ● | - | ● | ● |
| **INO** | Nitrosyl iodide | ● | ● | - | - | - |
| **INO₂** | Iodine nitrite | ● | ● | - | ● | ● |
| **INO₃** | Iodine nitrate | ● | ● | - | ● | ● |
| **CH₃I** | Iodomethane | ● | ● | ● | - | - |
| **CH₂I₂** | Diiodomethane | ● | ● | ● | - | - |
| **I₂O₂** | Iodooxy hypoiodite | ● | ● | - | ● | ● |
| **I₂O₃** | Iodo iodate | ● | ● | - | ● | ● |
| **I₂O₄** | Iodosyl iodate | ● | ● | - | ● | ● |
| **I_aer** | Aerosol iodine | - | ● | - | ● | ● |
| **Cl₂** | Dichlorine | ● | ● | - | ● | ● |
| **Cl** | Chlorine | ● | - | - | - | - |
| **HCl** | Hydrogen chloride | ● | ● | ● | ● | ● |
| **ClO** | Chlorine monoxide | ● | - | - | - | - |
| **ClNO₂** | Chlorine nitrite | ● | ● | ■ | ● | ● |
| **ClNO₃** | Chlorine nitrate | ● | ● | - | ● | ● |
| **ClOO** | Chlorine superoxide | ● | - | - | - | - |
| **OClO** | Chlorine dioxide | ● | - | - | - | - |
| **HOCl** | Hydrochlorous acid | ● | ● | - | ● | ● |
| **Cl₂O₂** | Dichlorine dioxide | ● | ● | - | - | - |
| **CH₃Cl** | Chloromethane | ● | ● | ● | - | - |
| **CH₂Cl₂** | Dichloromethane | ● | ● | ● | - | - |
| **CHCl₃** | Trichloromethane | ● | ● | ● | - | - |
| **CH₂IBr** | Bromoiodomethane | ● | ● | ● | - | - |
| **CH₂ICl** | Chloroiodomethane | ● | ● | ● | - | - |
| **IBr** | Iodine monobromide | ● | ● | ■ | ● | ● |
| **ICl** | Iodine monochloride | ● | ● | ■ | ● | ● |
| **BrCl** | Bromine monochloride | ● | ● | - | ● | ● |

[a] surface source ● or net chemical source from seasalts (dynamically represented in INCA) ■


**Table 2. Photolysis reactions of halogens included in the scheme.**

| Reaction | Quantum yield φ | Reference for absorption cross-section |
|---|---|---|
| Br₂ + hv → 2Br | 1 | Sander et al. (2011) |
| BrNO₂+ hv → Br + NO₂ | 1 | Sander et al. (2011) |
| BrONO₂ + hv → BrO + NO₂ | 0.15 | Sander et al. (2011) |
| BrONO₂ + hv → Br + NO₂ | 0.85 | Sander et al. (2011) |
| BrO + hv → Br | 1 | Sander et al. (2011) |
| CH₃Br + hv → Br | 1 | Sander et al. (2011) |
| CH₂Br₂ + hv → 2Br | 1 | Sander et al. (2011) |
| CHBr₃ + hv → 3Br | 1 | Sander et al. (2011) |
| HOBr + hv → Br + OH | 1 | Sander et al. (2011) |
| BrCl + hv → Br + Cl | 1 | Sander et al. (2011) |
| CH₂IBr + hv → I + Br | 1 | Sander et al. (2011) |
| IBr + hv → I + Br | 1 | Sander et al. (2011) |

| | | |
|---|---|---|
| CH$_2$I$_2$ + hv → 2I | 1 | Sander et al. (2011) |
| CH$_3$I + hv → I | 1 | Sander et al. (2011) |
| HOI + hv → I + OH | 1 | Sander et al. (2011) |
| INO$_3$ + hv → I + NO$_3$ | 1 | Sander et al. (2011) |
| I$_2$ + hv → 2I | 1 | Sander et al. (2011) |
| INO$_2$ + hv → I + NO$_2$ | 1 | Sander et al. (2011) |
| INO + hv → I +NO | 1 | Sander et al. (2011) |
| IO + hv → I | 0.91 | Sander et al. (2011) |
| OIO + hv → I + O$_2$ | 1 | Sander et al. (2011) |
| ICl + hv → I +Cl | 1 | Sander et al. (2011) |
| CH$_2$ICl + hv → I +Cl | 1 | Sander et al. (2011) |
| CH$_2$Cl$_2$ + hv → 2Cl | 1 | Sander et al. (2011) |
| CH$_3$Cl + hv → Cl + CH$_3$O$_2$ | 1 | Sander et al. (2011) |
| Cl$_2$O$_2$ + hv → Cl + ClOO | 1 | Sander et al. (2011) |
| Cl$_2$ + hv → 2Cl | 1 | Sander et al. (2011) |
| ClNO$_2$ + hv → Cl + NO$_2$ | 1 | Sander et al. (2011) |
| ClONO$_2$ + hv → Cl + NO$_2$ | Footnote #1 | Sander et al. (2011) |
| ClONO$_2$ + hv → ClO + NO$_2$ | Foonote #2 | Sander et al. (2011) |
| ClOO + hv → ClO + O$_2$ | 1 | Sander et al. (2011) |
| ClO + hv → Cl | 1 | Sander et al. (2011) |
| HOCl + hv → Cl + OH | 1 | Sander et al. (2011) |
| OClO + hv → ClO + O | 1 | Sander et al. (2011) |
| I$_2$O$_2$ + hv → I + OIO | 0.21 | Gómez Martin et al. (2005); Spietz et al. (2005) |
| I$_2$O$_3$ + hv → IO + OIO | 0.21 | Gómez Martin et al. (2005); Spietz et al. (2005) |
| I$_2$O$_4$ + hv → 2 OIO | 0.21 | Gómez Martin et al. (2005); Spietz et al. (2005) |

Footnotes of Table 2:
#1: $\varphi_1$ ($\lambda$ < 308 nm) = 0.6, $\varphi_1$ ($\lambda$=308-364 nm) = 7.143 x 10$^{-3}$ $\lambda$ (nm) – 1.60, $\varphi_1$($\lambda$ > 364 nm) = 1.0
#2: $\varphi_2$($\lambda$) = 1- $\varphi_1$($\lambda$)


**Table 3. Bimolecular halogen reactions included in the INCA scheme. The rate constant is calculated using a standard Arrhenius expression A x exp( −Ea /RT).**

| Reaction | $A_0$ (molécules$^{-2}$cm$^6$s$^{-1}$) | $-E_a/R$ (K) | Reference |
|---|---|---|---|
| Cl + CH$_3$O$_2$ → ClO + CH$_2$O + HO$_2$ | 1.60 x 10$^{-10}$ | | Sander et al. (2011) |
| Cl + CH$_3$OOH → HCl + CH$_3$O$_2$ | 5.70 x 10$^{-11}$ | | Sander et al. (2011) |
| Cl + C$_2$H$_6$ → HCl + C$_2$H$_5$O$_2$ | 7.20 x 10$^{-11}$ | -70 | Sander et al. (2011) |
| Cl + C$_2$H$_5$O$_2$ → ClO + HO$_2$ + CH$_3$CHO | 7.40 x 10$^{-11}$ | | Sander et al. (2011) |
| Cl + C$_2$H$_5$OH → HCl + CH$_3$CHO | 9.60 x 10$^{-11}$ | | Sander et al. (2011) |
| Cl + CH$_3$COOH → HCl + CH$_3$O$_2$ + CO$_2$ | 2.80 x 10$^{-14}$ | | Sander et al. (2011) |
| Cl + C$_3$H$_8$ → HCl + C$_3$H$_7$O$_2$ | 7.85 x 10$^{-11}$ | -80 | Sander et al. (2011) |
| Cl + C$_3$H$_8$ → HCl + PROPAO$_2$ | 6.54 x 10$^{-11}$ | | Sander et al. (2011) |
| Cl + CH$_3$COCH$_3$ → HCl + PROPAO$_2$ | 7.70 x 10$^{-11}$ | | Sander et al. (2011) |
| Cl + ISOP → HCl + ISOPO$_2$ | 7.70 x 10$^{-11}$ | 500 | Sander et al. (2011) |
| Cl + CH$_3$OH → HCl + CH$_2$O + HO$_2$ | 5.50 x 10$^{-11}$ | | Sander et al. (2011) |
| Cl + ALKAN → HCl + ALKANO$_2$ | 2.05 x 10$^{-10}$ | | Atkinson et al. (2006) |
| Cl + C$_3$H$_6$ → HCl + PROPEO$_2$ | 3.60 x 10$^{-12}$ | | Atkinson et al. (2006) |
| Cl + CH$_3$Cl → CO + 2 HCl + HO$_2$ | 2.17 x 10$^{-11}$ | -1130 | Sander et al. (2011) |
| Cl + H$_2$O$_2$ → HCl + HO$_2$ | 1.10 x 10$^{-11}$ | -980 | Sander et al. (2011) |
| Cl + HO$_2$ → HCl + O$_2$ | 1.40 x 10$^{-11}$ | 270 | Sander et al. (2011) |
| Cl + HO$_2$ → ClO + OH | 3.60 x 10$^{-11}$ | -375 | Sander et al. (2011) |
| Cl + O$_3$ → ClO + O$_2$ | 2.30 x 10$^{-11}$ | -200 | Sander et al. (2011) |
| Cl + ClNO$_3$ → Cl$_2$ + NO$_3$ | 6.50 x 10$^{-12}$ | 135 | Sander et al. (2011) |
| ClO + ClO → Cl$_2$ + O$_2$ | 1.00 x 10$^{-12}$ | -1590 | Sander et al. (2011) |
| ClO + ClO → OClO + Cl | 3.50 x 10$^{-13}$ | -1370 | Sander et al. (2011) |
| ClO + ClO → Cl + ClOO | 3.00 x 10$^{-11}$ | -2450 | Sander et al. (2011) |
| ClO + HO$_2$ → O$_2$ + HOCl | 2.60 x 10$^{-12}$ | 290 | Sander et al. (2011) |
| ClO + NO → Cl + NO$_2$ | 6.40 x 10$^{-12}$ | 290 | Sander et al. (2011) |
| ClOO + Cl → 2 ClO | 1.20 x 10$^{-11}$ | | Sander et al. (2011) |
| ClOO + Cl → Cl$_2$ + O$_2$ | 2.30 x 10$^{-10}$ | | Sander et al. (2011) |
| ClO + CH$_3$O$_2$ → ClOO + HO$_2$ + CH$_2$O | 3.30 x 10$^{-12}$ | -115 | Sander et al. (2011) |
| OH + CH$_3$Cl → Cl + HO$_2$ + H$_2$O | 3.90 x 10$^{-12}$ | -1411 | Sander et al. (2011) |

| Reaction | | $A$ | $E/R$ | Reference |
|---|---|---|---|---|
| $OH + CH_2Cl_2 \rightarrow 2\,Cl + HO_2 + H_2O$ | | $1.90 \times 10^{-12}$ | -870 | Sander et al. (2011) |
| $OH + CHCl_3 \rightarrow 3\,Cl + HO_2 + H_2O$ | | $2.20 \times 10^{-12}$ | -920 | Sander et al. (2011) |
| $OH + Cl_2 \rightarrow HOCl + Cl$ | | $2.60 \times 10^{-12}$ | -1100 | Sander et al. (2011) |
| $OH + Cl_2O_2 \rightarrow HOCl + ClOO$ | | $6.00 \times 10^{-13}$ | 670 | Sander et al. (2011) |
| $OH + ClNO_2 \rightarrow HOCl + NO_2$ | | $2.40 \times 10^{-12}$ | -12 50 | Sander et al. (2011) |
| $OH + ClNO_3 \rightarrow HOCl + NO_3$ | | $1.20 \times 10^{-12}$ | -330 | Sander et al. (2011) |
| $OH + ClO \rightarrow HCl + O_2$ | | $6.00 \times 10^{-13}$ | 230 | Sander et al. (2011) |
| $OH + ClO \rightarrow HO_2 + Cl$ | | $7.40 \times 10^{-12}$ | 270 | Sander et al. (2011) |
| $OH + HCl \rightarrow H_2O + Cl$ | | $1.80 \times 10^{-12}$ | -250 | Sander et al. (2011) |
| $OH + HOCl \rightarrow H_2O + ClO$ | | $3.00 \times 10^{-12}$ | -500 | Sander et al. (2011) |
| $OH + OClO \rightarrow HOCl + O_2$ | | $1.50 \times 10^{-12}$ | 600 | Sander et al. (2011) |
| $Cl + CH_4 \rightarrow HCl + CH_3O_2$ | | $9.60 \times 10^{-12}$ | -1360 | Atkinson et al. (2004) |
| $Cl + C_2H_4 \rightarrow HCl + C_2H_5O_2$ | | $1.00 \times 10^{-10}$ | | Lurmann et al. (1986) |
| $Cl + CH_2O \rightarrow HCl + HO_2 + CO$ | | $8.10 \times 10^{-11}$ | -30 | Sander et al. (2003) |
| $Cl + PAN \rightarrow HCl + CH_2O + NO_3$ | | $1.00 \times 10^{-14}$ | | Sander et al. (2003) |
| $Cl + HNO_3 \rightarrow HCl + NO_2$ | | $1.00 \times 10^{-16}$ | | Sander et al. (2003) |
| $Br + O_3 \rightarrow BrO + O_2$ | | $1.60 \times 10^{-11}$ | -780 | Sander et al. (2011) |
| $Br + HO_2 \rightarrow HBr + O_2$ | | $4.80 \times 10^{-12}$ | -310 | Sander et al. (2011) |
| $Br + CH_2O \rightarrow HO_2 + CO + HBr$ | | $1.70 \times 10^{-11}$ | -800 | Sander et al. (2011) |
| $Br + C_2H_6 \rightarrow C_2H_5O_2 + HBr$ | | $2.36 \times 10^{-10}$ | -6411 | Seakins et al. (1992) |
| $Br + C_3H_8 \rightarrow C_3H_7O_2 + HBr$ | | $8.77 \times 10^{-11}$ | -4330 | Seakins et al. (1992) |
| $Br + CH_3CHO \rightarrow CH_3CO_3 + HBr$ | | $1.30 \times 10^{-11}$ | -360 | Atkinson et al. (2007) |
| $Br + CH_3COCH_3 \rightarrow PROPAO_2 + HBr$ | | $1.66 \times 10^{-10}$ | -7000 | King et al. (1970) |
| $Br + C_3H_6 \rightarrow PROPEO_2 + HBr$ | | $3.60 \times 10^{-12}$ | | Atkinson et al. (2006) |
| $Br + ALKEN \rightarrow ALKENO_2 + HBr$ | | $3.60 \times 10^{-12}$ | | Atkinson et al. (2006) |
| $Br + BrNO_3 \rightarrow Br_2 + NO_3$ | | $4.90 \times 10^{-11}$ | | Orlando and Tyndall (1996) |
| $Br + NO_3 \rightarrow BrO + NO_2$ | | $1.60 \times 10^{-11}$ | | Sander et al. (2011) |
| $HBr + OH \rightarrow Br + H_2O$ | | $5.50 \times 10^{-12}$ | 200 | Sander et al. (2011) |
| $BrO + OH \rightarrow Br + HO_2$ | | $1.70 \times 10^{-11}$ | 250 | Sander et al. (2011) |
| $BrO + HO_2 \rightarrow HOBr + O_2$ | | $4.50 \times 10^{-12}$ | 460 | Sander et al. (2011) |
| $BrO + NO \rightarrow Br + NO_2$ | | $8.80 \times 10^{-12}$ | 260 | Sander et al. (2011) |
| $BrO + BrO \rightarrow 2\,Br + O_2$ | | $2.40 \times 10^{-12}$ | 40 | Sander et al. (2011) |
| $BrO + BrO \rightarrow Br_2 + O_2$ | | $2.80 \times 10^{-14}$ | 860 | Sander et al. (2011) |
| $Br_2 + OH \rightarrow HOBr + Br$ | | $2.10 \times 10^{-11}$ | 240 | Sander et al. (2011) |
| $CHBr_3 + OH \rightarrow 3\,Br + CO$ | | $1.35 \times 10^{-12}$ | -600 | Sander et al. (2011) |
| $CH_2Br_2 + OH \rightarrow 2\,Br + CO$ | | $2.00 \times 10^{-12}$ | -840 | Sander et al. (2011) |
| $CH_3Br + OH \rightarrow Br + CO$ | | $2.35 \times 10^{-12}$ | -1300 | Sander et al. (2011) |
| $I + O_3 \rightarrow IO + O_2$ | | $2.10 \times 10^{-10}$ | -830 | Atkinson et al. (2007) |
| $I + HO_2 \rightarrow HI + O_2$ | | $1.50 \times 10^{-11}$ | -1090 | Sander et al. (2011) |
| $I_2 + OH \rightarrow HOI + I$ | | $2.10 \times 10^{-10}$ | | Atkinson et al. (2007) |
| $HI + OH \rightarrow I + H_2O$ | | $1.60 \times 10^{-11}$ | 440 | Atkinson et al. (2007) |
| $HOI + OH \rightarrow IO + H_2O$ | | $5.00 \times 10^{-12}$ | | Riffault et al. (2005) |
| $IO + HO_2 \rightarrow HOI + O_2$ | | $1.40 \times 10^{-11}$ | 540 | Atkinson et al. (2007) |
| $IO + NO \rightarrow I + NO_2$ | | $7.15 \times 10^{-12}$ | 300 | Atkinson et al. (2007) |
| $CH_3I + OH \rightarrow H_2O + I$ | | $4.30 \times 10^{-12}$ | -112 0 | Atkinson et al. (2008) |
| $INO + INO \rightarrow I_2 + 2\,NO$ | | $8.40 \times 10^{-11}$ | -2620 | Atkinson et al. (2007) |
| $INO_2 + INO_2 \rightarrow I_2 + 2\,NO_2$ | | $4.70 \times 10^{-12}$ | -1670 | Atkinson et al. (2007) |
| $I_2 + NO_3 \rightarrow I + INO_3$ | | $1.50 \times 10^{-12}$ | | Atkinson et al. (2007) |
| $INO_3 + I \rightarrow I_2 + NO_3$ | | $9.10 \times 10^{-11}$ | -146 | Kaltsoyannis and Plane (2008 |
| $OIO + OIO \rightarrow I_2O_4$ | | $1.50 \times 10^{-10}$ | | Gómez Martin et al. (2007) |
| $OIO + NO \rightarrow NO_2 + IO$ | | $1.10 \times 10^{-12}$ | 542 | Atkinson et al. (2007) |

| Reaction | A | | Reference |
|---|---|---|---|
| $IO + IO \rightarrow I + OIO$ | $2.16 \times 10^{-11}$ | 180 | Atkinson et al. (2007) |
| $IO + IO \rightarrow I_2O_2$ | $3.24 \times 10^{-11}$ | 180 | Atkinson et al. (2007) |
| $IO + OIO \rightarrow I_2O_3$ | $1.50 \times 10^{-10}$ | | Gómez Martin et al. (2007) |
| $I_2O_2 \rightarrow IO + IO$ | $1.00 \times 10^{+12}$ | -9770 | Ordóñez et al. (2012) |
| $I_2O_2 \rightarrow OIO + I$ | $2.50 \times 10^{+14}$ | -9770 | Ordóñez et al. (2012) |
| $I_2O_4 \rightarrow 2\ OIO$ | $3.80 \times 10^{-02}$ | | Kaltsoyannis and Plane. (2008) |
| $INO_2 \rightarrow I + NO_2$ | $9.94 \times 10^{+17}$ | -11859 | McFiggans et al. (2000) |
| $INO_3 \rightarrow IO + NO_2$ | $2.10 \times 10^{+15}$ | -13670 | Kaltsoyannis and Plane. (2008) |
| $IO + ClO \rightarrow I + OClO$ | $2.59 \times 10^{-11}$ | 280 | Atkinson et al. (2007) |
| $IO + ClO \rightarrow I + Cl + O_2$ | $1.18 \times 10^{-12}$ | 280 | Atkinson et al. (2007) |
| $IO + ClO \rightarrow ICl + O_2$ | $9.40 \times 10^{-13}$ | 280 | Atkinson et al. (2007) |
| $I + BrO \rightarrow IO + Br$ | $1.20 \times 10^{-11}$ | | Sander et al. (2011) |
| $IO + Br \rightarrow I + BrO$ | $2.70 \times 10^{-11}$ | | Bedjanian et al. (1997) |
| $IO + BrO \rightarrow Br + I + O_2$ | $3.00 \times 10^{-12}$ | 510 | Atkinson et al. (2007) |
| $IO + BrO \rightarrow Br + OIO$ | $1.20 \times 10^{-11}$ | 510 | Atkinson et al. (2007) |
| $ClO + BrO \rightarrow OClO + Br$ | $1.60 \times 10^{-12}$ | 430 | Atkinson et al. (2004) |
| $ClO + BrO \rightarrow Br + Cl + O_2$ | $2.90 \times 10^{-12}$ | 220 | Atkinson et al. (2004) |
| $ClO + BrO \rightarrow BrCl + O_2$ | $5.80 \times 10^{-13}$ | 170 | Atkinson et al. (2004) |

Footnote of Table 3: The reactions come from GEOS-Chem (Parrella et al., 2012; Eastham et al., 2014; Schmidt et al., 2016; Sherwen et al., 2016a; Sherwen et al., 2016b), TOMCAT (Hossaini et al., 2016) and Thomas et al., 2011. Lumped species are detailed in Folberth et al., 2006.


**Table 4. Termolecular halogen reactions included in the INCA scheme.**

| Termolecular reaction | $A_0$ (molecules$^{-2}$cm$^6$s$^{-1}$) | x | $k_\infty$ | m | Fc | Reference |
|---|---|---|---|---|---|---|
| $Cl + O_2 + M \rightarrow ClOO$ | $2.20 \times 10^{-33}$ | 0 | $1.80 \times 10^{-10}$ | 3.1 | 0.6 | Sander et al. (2011) |
| $ClO + ClO + M \rightarrow Cl_2O_2$ | $1.60 \times 10^{-21}$ | 2 | $3.0 \times 10^{-12}$ | 4.5 | 0.6 | Sander et al. (2011) |
| $ClO + NO_2 + M \rightarrow ClNO_3$ | $1.80 \times 10^{-31}$ | 3.4 | $1.50 \times 10^{-11}$ | 1.9 | 0.6 | Sander et al. (2011) |
| $ClOO + M \rightarrow Cl + O_2$ | $3.30 \times 10^{-9}$ | 0 | $2.73 \times 10^{14}$ | 3.1 | 0.6 | Sander et al. (2011) |
| $Cl_2O_2 + M \rightarrow Cl + O_2$ | $9.30 \times 10^{-6}$ | 2 | $1.74 \times 10^{15}$ | 4.5 | 0.6 | Sander et al. (2011) |
| $Cl + C_3H_6 + M \rightarrow ALKANO_2$ | $4.0 \times 10^{-28}$ | 0 | $2.80 \times 10^{-10}$ | - | 0.6 | Atkinson et al. (2006) |
| $Br + NO_2 + M \rightarrow BrNO_2$ | $4.20 \times 10^{-31}$ | 2.4 | $2.70 \times 10^{-11}$ | - | 0.6 | Sander et al. (2011) |
| $BrO + NO_2 + M \rightarrow BrNO_3$ | $5.20 \times 10^{-31}$ | 3.2 | $6.90 \times 10^{-12}$ | - | 0.6 | Sander et al. (2011) |
| $I + NO + M \rightarrow INO$ | $1.80 \times 10^{-32}$ | 1 | $1.70 \times 10^{-11}$ | - | 0.6 | Atkinson et al.(2007) |
| $I + NO_2 + M \rightarrow INO_2$ | $3.0 \times 10^{-31}$ | 1 | $6.60 \times 10^{-11}$ | - | 0.63 | Atkinson et al.(2007) |
| $IO + NO2 + M \rightarrow INO_3$ | $7.70 \times 10^{-31}$ | 5 | $1.60 \times 10^{-11}$ | - | 0.4 | Atkinson et al.(2007) |

Footnote of Table 4: The reactions come from previous updates to halogen chemistry in GEOS-Chem (Parrella et al., 2012; Eastham et al., 2014; Schmidt et al., 2016; Sherwen et al., 2016a; Sherwen et al., 2016b). The lower pressure limit rate (k0) is given by $A_0(\frac{300}{T})^x$. The high pressure limit is given by k∞. Fc characterizes the fall off curve of the reaction as described by Atkinson et al. (2007).

**Table 5. Halogen multiphase reactions and reactive uptake coefficients (γ).**

| Reaction | Reactive uptake coefficient on sea salt (γ) | Reactive uptake coefficient on sulfate aerosols (γ) | *Reference* |
|---|---|---|---|
| $HOBr + HBr \rightarrow Br_2 + H_2O$ [c] | 0.2 | 0.2 | Parrella et al. (2012) |
| $HOBr + HCl \rightarrow BrCl + H_2O$ [c] | 0.2 | 0.2 | Sander et al. (2011) |
| $ClNO_3 + HBr \rightarrow BrCl + HNO_3$ [c] | 0.2 | 0.2 | Compromise between Sander et al. (2011) and Badia et al. (2019) |
| $ClNO_3 \rightarrow HOCl + HNO_3$ | 0.001[a] 0.01[b] | 0.001[a] 0.01[b] | Badia et al. 2019 |
| $BrNO_3 \rightarrow HOBr + HNO_3$ | 0.03[a] | 0.03[a] | Badia et al. 2019 |

| | 0.8[b] | 0.8[b] | |
|---|---|---|---|
| $I_2O_x \rightarrow 2\ I_{aer.}$ [d] | 0.02 | $2.10^{-2}$ | Sherwen et al. 2016 |
| $HI \rightarrow I_{aer}$ | 0.1 | - | Sherwen et al. 2016 |
| $HOI \rightarrow 0.85\ ICl + 0.15\ IBr + HNO_3$ | 0.01 | - | Sherwen et al. 2016 |
| $INO_3 \rightarrow 0.85\ ICl + 0.15\ IBr + HNO_3$ | 0.01 | - | Sherwen et al. 2016 |
| $INO_2 \rightarrow 0.85\ ICl + 0.15\ IBr + HNO_3$ | 0.02 | - | Sherwen et al. 2016 |
| $N_2O_5 \rightarrow 1.5\ HNO_3 + 0.5\ ClNO_2$ | $3.10^{-2}$ | - | Hossaini et al. 2016 |

[a]Uptake coefficient for moderate temperatures.

[b]Uptake coefficient for cold temperatures.

[c]For second order reaction, rate constants are calculated by assuming that the first reactant is limiting thus, the first-order rate constant is divided by the concentrations of the adsorbed species

[d] $I_2O_x$ represent $I_2O_2$, $I_2O_3$ and $I_2O_4$

## 2.2 Simulations

For these developments, LMDZ-INCA is used with a 3.75° longitude× 1.9° latitude resolution over 39 vertical layers (up to 70km with 15 levels in the stratosphere). This resolution, coarser than the resolution now commonly used in the CMIP6 model

exercise, was the standard LMDZ resolution for CMIP5 (Hourdin et al. 2012) and allows faster simulations necessary in model development. The LMDZ wind fields and sea surface temperatures are nudged on ECMWF reanalysis for the year 2010 in all simulations. The aerosol distribution and monthly variations used for heterogeneous reactions calculations are specified from climatologies precomputed with another version of the same model but with fully interactive calculations of aerosols. In this version, the stratospheric composition is prescribed. All simulations cover 1 year (2010), and follow a spin-up period of one

year. Three types of simulations have been performed to simulate (i) present-day conditions, (ii) preindustrial conditions, and (iii) sensitivity to ozone precursors. The surface emissions of NOx, CO, and NMVOCs and $CH_4$ concentrations considered in the present-day and preindustrial simulations are presented in Table S2. For sensitivity tests, simulations with individual emission perturbations of photooxidant precursors (-20%) of $NO_x$, CO and NMVOCs and concentration reduction (-20%) of $CH_4$, were performed. For each type of simulation, two versions of the model are applied: one accounting for the halogen

chemistry (called Halo simulations hereafter) and a reference simulation without halogen chemistry called "NoHalo".

### 2.2.1 Present-day set-up

Primary halogenated compounds can originate from both inorganic and organic sources. Inorganic species are reactive species emitted by different sources (e.g. oceans, sea salts), but also result from atmospheric reactions (Simpson et al., 2015). Organic species are only of primary origin and are photooxidized in the atmosphere to form halogen radicals (Simpson et al., 2015).

Table 6 shows the emissions of halogenated compounds as considered in INCA. Organo-bromines (e.g. $CHBr_3$, $CH_2Br_2$), organo-iodines (e.g. $CH_3I$, $CH_2I_2$) and interhalogen species (e.g. $CH_2IBr$, $CH_2ICl$) are those used by the CAM-Chem model (Ordóñez et al., 2012).

Sources of organic chlorine are diverse, mainly originating from biological activity on the surface of the oceans and from forest fires. The emissions of these chlorocarbons are detailed in Table S2. In INCA, inorganic chlorine, HCl - the major reservoir of chlorine in the atmosphere - is considered as being emitted by sea salts (90 000 Gg Cl.yr$^{-1}$) and by forest fires (6400 Gg

Cl.yr$^{-1}$) with total based on Lobert et al., 1999 and Hossaini et al., 2016. Note that more comprehensive but hard to implement approaches relying on pH dependent parameterization of the dechlorination of sea salts have been recently implemented in

GEOS-Chem. HCl emissions from industrial sources (McCulloch et al., 1999) are neglected as recommended by Wang et al. (2019), to avoid overestimating HCl in urban areas. ClNO$_2$ is emitted by the heterogeneous reaction of N$_2$O$_5$ on sea salts and produces 2217 Gg Cl.yr$^{-1}$.

CH$_3$Br emissions are from oceanic origin (40%) and from agriculture (60%) (Barker et al., 2016) and represent 91 Gg of Br.yr$^{-1}$ (Schmidt et al. (2016). IBr and ICl originate from the heterogeneous reaction of HOI, INO$_2$ and INO$_3$ on sea salt aerosols (Table 5) as considered in other models (Saiz-Lopez et al., 2014; Sherwen et al., 2016b), but without accounting for pH dependence (McFiggans et al., 2000).

The inorganic iodine distribution of emissions (HOI and I$_2$) are prescribed from a climatology computed by the CAM-Chem model considering sea-air fluxes as a function of the oxidation of aqueous iodide by atmospheric ozone on the ocean surface parameterized by Carpenter et al., 2013; MacDonald et al., 2011 and which depends on the ozone deposition to the ocean surface, the sea surface temperature and the wind speed (Prados-Roman et al., 2015b).

### 2.2.2 Preindustrial set-up

The aim of this simulation is to explore how the sensitivity of ozone to anthropogenic emissions perturbations changes since the preindustrial period (a present day climate is thus considered). Only emissions from natural sources are included for the simulation of the preindustrial troposphere, following the approach used by other modelling teams to quantify this perturbation. Emissions of halogenated compounds from biomass burning are reduced to 10% of their current values (Wang and Jacob, 1998). The methane concentration is set at 700 ppbv (IPCC, 2014). Emissions of biogenic volatile organic compounds are kept at their present-day level due to uncertainties in the effect of their drivers over the past century and the resulting high uncertainty regarding the quantification of change in their emissions (Szopa et al. 2021).

Organo-iodine emissions are almost unchanged between the PI and PD. For organo-bromines, preindustrial emissions of CHBr$_3$ and CH$_2$Br$_2$ are equivalent to the PD ones, following the methodology of Parrella et al. (2012). The concentration of CH$_3$Br is set to ~ 5 ppbv is considered in agreement with the measurements from ice cores (Saltzman et al., 2004). Preindustrial emissions of CH$_3$Cl, CHCl$_3$ and CH$_2$Cl$_2$ are from (Carpenter et al., 2014; Hu, 2012; Montzka et al., 2011; Worton et al., 2006). HOI and I$_2$ emissions, pre-computed by CAM-Chem, are ~ 40% lower than today because they are sensitive to atmospheric ozone content which was lower in the pre-industrial era (Prados-Roman et al., 2015a; Sherwen et al., 2017). Global emissions for each primary halogenated compound are reported in Table 6.

**Table 6: Emission of halogen gases for preindustrial and present-day simulations.**

| Compounds | I (Gg I.an$^{-1}$) | | Br (Gg Br.an$^{-1}$) | | Cl (Gg Cl.an$^{-1}$) | |
|---|---|---|---|---|---|---|
| | Preindustrial | Present | Preindustrial | Present | Preindustrial | Present |
| CH$_3$X | 209 | 219 | 36 | 91 | 2081 | 2166 |
| CH$_2$X$_2$ | 108 | 108 | 62 | 62 | 199 | 628 |
| CHX$_3$ | - | - | 506 | 506 | 236 | 315 |
| HOI | 1361 | 2448 | - | - | - | - |
| I$_2$ | 34 | 59 | - | - | - | - |
| ClNO$_2$ | - | - | - | - | 169 | 2217 |
| CH$_2$IBr | 50 | 50 | 32 | 32 | - | - |
| CH$_2$ICl | 168 | 168 | - | - | 47 | 47 |
| IBr, ICl | - | - | 495 | 799 | 1244 | 2008 |
| HCl | - | - | - | - | 96429 | 96429 |
| Total | **1930** | **3052** | **1131** | **1490** | **100406** | **103810** |

Footnote of Table 6: Sources of organic species (CH$_3$X, CH$_2$X$_2$, CHX$_3$) are shown in terms of emissions. I$_2$ and HOI are the inorganic oceanic source that depends on surface ozone (Carpenter et al., 2013). IBr and ICl are released following gaseous iodine uptake on sea salt. ClNO$_2$ is emitted after absorption of N$_2$O$_5$ on sea salt.

## 3 Evaluation of the representation of the halogen-driven ozone sinks

We first assess the ozone sensitivity to halogen chemistry in LMDZ-INCA, then we evaluate the performance of the model in simulating the climatology of halogen species which are important for the global ozone loss. In order to do so, model outputs for compounds of interest in this chemistry are compared to a range of halogen observations and simulations from other models with halogen chemistry. An analysis of the model tropospheric budgets of key halogen species is also performed.

### 3.1 Impact on ozone loss in present-day

Table 7 presents the ozone budget in LMDZ-INCA with and without consideration of halogen chemistry. Overall, when halogens are considered, the tropospheric ozone burden is reduced by 22% for present-day conditions which is in the 5 to 25% range reported from other models which include Cl, Br and I chemistry (Saiz-Lopez et al., 2012; Saiz-Lopez et al., 2015, 2012a; Sherwen et al., 2016b; Badia et al., 2019, Badia et al. 2021). Over the global troposphere, it is associated to a decrease of the annual-average ozone mixing ratio (mass-weighted) of ozone of 9.7 ppbv (compared to 9.4 ppbv in GEOS-Chem, Sherwen et al., 2016b). Over the tropospheric tropical column, the annual-average ozone mixing ratio (mass-weighted) decreases by 8 ppbv (21%) which is twice as large as the relative change reported in Saiz-Lopez et al. (2014). Surface ozone averaged over continental areas undergoes a decrease of 6 ppbv (23%) while surface ozone in oceanic areas, where the majority of primary halogenated compounds are emitted, decreases by 7.7 ppbv (32%) (range reported in other studies is 20-33%; Saiz-Lopez et al., 2014; Long et al., 2014; Prados-Roman et al., 2015a; Sherwen et al., 2016b).

Figure 2 shows the absolute changes in surface ozone and relative change in zonally-averaged ozone in a simulation with halogen chemistry relative to a simulation without it. Significant relative losses are found in the Southern Hemisphere, which are due to the strong emissions of halogenated species from the ocean in this region (Long et al., 2014; Saiz-lopez et al., 2012; Schmidt et al., 2016; Sherwen et al., 2016b, 2016a) and the smaller absolute abundance of ozone. Most of the change in total tropospheric ozone concentration occurs in the free troposphere between 350 hPa and 900 hPa (63%). This result is similar in magnitude to the relative change reports in GEOS-Chem (65%; Sherwen et al., 2016b) and CAM-Chem (65%; Iglesias-Suarez et al., 2020).

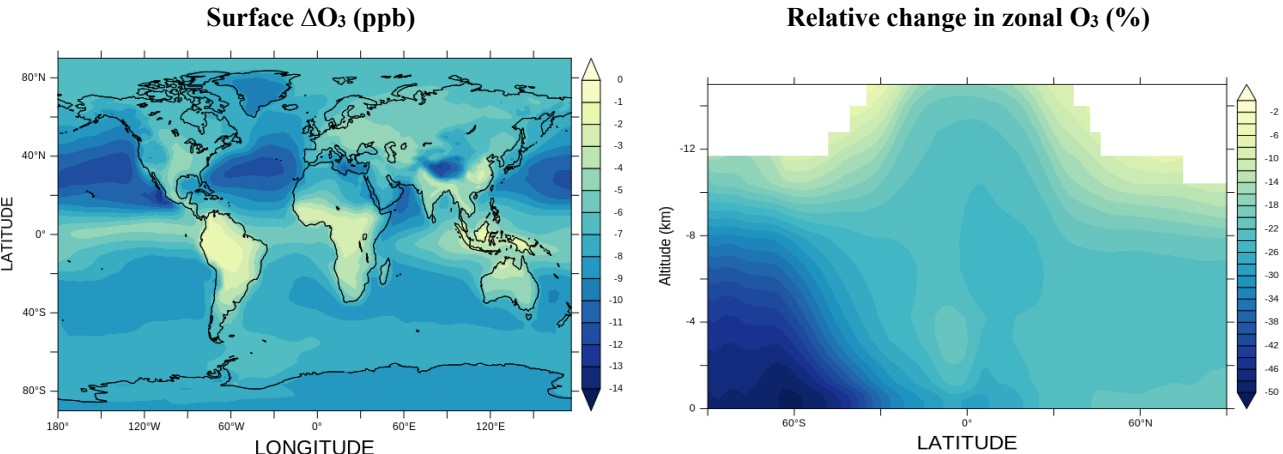

**Figure 2: Surface ozone changes (ppbv) and relative change in zonal tropospheric ozone (%) between simulations with and without halogen chemistry.**

The annual production of ozone decreases by 4.7% when the chemistry of halogens is considered. This decrease is due to a reduction in $NO_x$ concentrations of 32.8% (Table 7) due to the hydrolysis of $XNO_3$ (X = Cl, Br, I) leading to deposition losses, in accordance with previous studies (Saiz-Lopez et al., 2012b; Long et al., 2014; von Glasow et al., 2004; Parrella et al., 2012; Schmidt et al., 2016, Sherwen et al., 2016b).

The vertical profile of the different contributions to chemical ozone loss is shown Figure 3. The processes leading to ozone loss differ considerably between simulations with and without halogenated compounds (Table 7). Halogens account for nearly ~22% of the ozone loss in the boundary layer in the halogen simulation, as shown in Fig. 3. Halogens represent 55% of the Ox (Ox=Odd Oxygen) loss in the upper troposphere (350 hPa > pressure > tropopause) while this value is about 33% and 40% in GEOS-Chem with the iodine+bromine, and the iodine+bromine+chlorine chemistry, respectively. Such differences could be partly due to differences in convection schemes which induce different uplifts of IO in the upper troposphere resulting in different IO concentrations (see section 3.1.3 below) but also to different representations of the stratospheric composition/chemistry between models.

When chemistry of halogenated compounds is considered, the total annual chemical ozone loss decreases by 2.4% (Table 7). The relative contributions of different halogens to ozone loss are very similar to those of GEOS-Chem in Sherwen et al. (2016a). The loss of ozone by chlorine, bromine and iodine represents 0.5%, 4.7% and 18% of the total losses, respectively; thus, halogens represent 23.2% of the total loss of ozone. The sum of the ozone loss caused by the halogens is 984 Tg.yr$^{-1}$. Iodine is responsible for 78% of that halogen-mediated loss, while bromine accounts for 19%, which is close to the values reported by Iglesias-Suarez et al., 2020 (80%, 16% respectively). Iodine is responsible for an overall tropospheric ozone loss of 762 Tg.yr$^{-1}$ (comparable to the 748 Tg.yr$^{-1}$ reported by Sherwen et al., 2016a). Much of this loss arises from the photolysis of HOI after its production from the reaction of IO with HO$_2$ and the photolysis of OIO after its production from reactions implicating IO and I$_x$O$_y$. Bromine is responsible for an overall tropospheric loss of 187 Tg.yr$^{-1}$ which mostly arises from the photolysis of HOBr after its production from the reaction of BrO with HO$_2$.

**Global vertical ozone loss in LMDZ-INCA without halogen chemistry**       **Global vertical ozone loss in LMDZ-INCA with halogen chemistry**

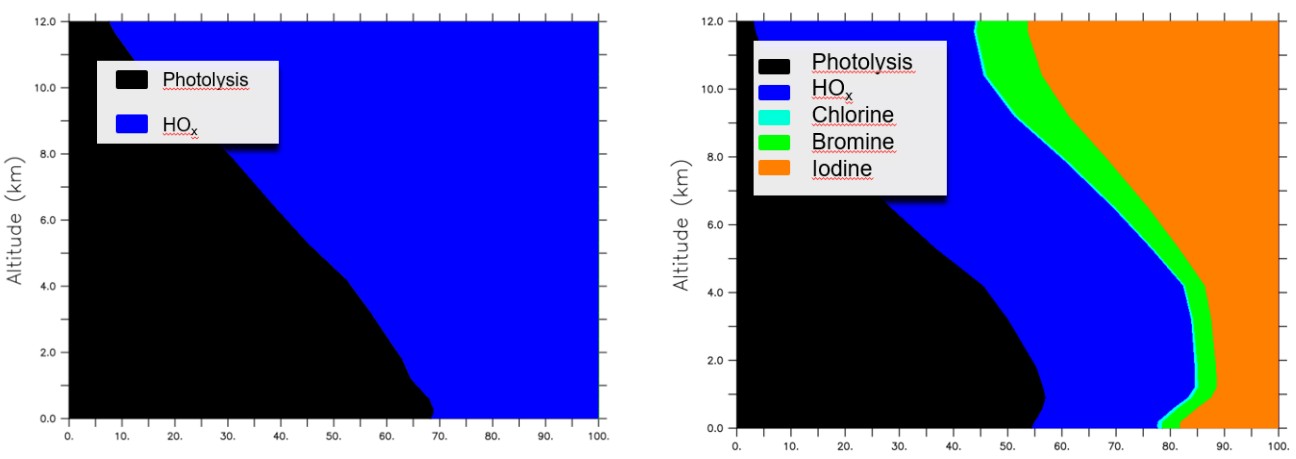

**Figure 3: Contribution of the chemical ozone sinks as a function of the altitude without (left) and with (right) consideration of the chemistry of halogens in LMDZ-INCA.**

The inclusion of halogens leads to an overall decrease in the net production of tropospheric ozone (PO$_3$-LO$_3$) of 20% (126 Tg.yr$^{-1}$). Consequently, the lifetime of ozone (=tropospheric burden/losses) is significantly decreased by 3.2 days and reaches 17.9 days.

**Table 7: Comparison of global tropospheric O$_x$ budgets of "Halo" and "NoHalo" simulations in INCA for the PI and PD.**

|  | Preindustrial Halo | Preindustrial NoHalo | Present day Halo | Present day NoHalo |
|---|---|---|---|---|
| **Ozone burden (Tg)** | 170.7 | 218.9 | 249.9 | 317 |
| **Ozone chemical sources (Tg.yr$^{-1}$)** | | | | |
| **NO + HO$_2$** | 1459 | 1554 | 2954 | 3190 |

| | | | | |
|---|---|---|---|---|
| NO + CH₃O₂ | 474.9 | 477.2 | 1134 | 1159 |

| | | | | |
|---|---|---|---|---|
| NO + CH$_3$O$_2$ | 474.9 | 477.2 | 1134 | 1159 |
| NO + RO$_2$ | 380.7 | 355.5 | 625.8 | 596.7 |
| **Total chemical ozone sources (PO$_3$)** | **2314** | **2387** | **4714** | **4946** |
| **Ozone chemical sinks (Tg.yr$^{-1}$)** | | | | |
| O$_3$ + H$_2$O → 2 OH + O$_2$ | 1099 | 1497 | 1905 | 2460 |
| O$_3$ + HO$_2$ → OH + O$_2$ | 431.5 | 648 | 917 | 1295 |
| O$_3$ + OH → HO$_2$ + O$_2$ | 207.1 | 297.8 | 349.4 | 488.7 |
| Bromine O$_3$ sinks | 155.3 | - | 186.9 | - |
| Iodine O$_3$ sinks | 536.9 | - | 762.3 | - |
| Chlorine O$_3$ sinks | 23.51 | - | 22.16 | |
| Other O$_3$ sinks | 78.06 | 85.11 | 88.5 | 92.87 |
| **Total chemical ozone sink (LO$_3$)** | **2531** | **2528** | **4231** | **4337** |
| Ozone PO$_3$-LO$_3$ (Tg.yr$^{-1}$) | -216.6 | -141 | 483 | 609.1 |
| Ozone dry deposition (Tg.yr$^{-1}$) | 403.4 | 593.4 | 875.5 | 1139 |
| Ozone lifetime (days) | 21.2 | 25.6 | 17.9 | 21.1 |
| Ozone STE (LO$_3$+Dry dep-PO$_3$) (Tg.yr$^{-1}$) | 620 | 734.5 | 392.4 | 530.2 |

## 3.2 Present-day budgets for halogenated species in LMDZ-INCA

In order to reflect the role played by the halogenated species in their respective cycle, inorganic halogens are presented hereafter under the name X$_y$ which can be divided into HX (the acidic forms of halogenated compounds) and X* (composed of the nocturnal reservoirs of reactive halogenated species and the reactive halogenated species). HX in LMDZ-INCA takes part in heterogeneous bimolecular reactions. In CAM-Chem and GEOS-Chem, the representation of this chemistry is more detailed: HX can be incorporated in droplets or particles and dissociate (Fernandez et al., 2014b; Ordóñez et al., 2012; Prados-Roman et al., 2015a; Saiz-Lopez et al., 2012b, 2014, 2015, Sherwen et al. 2016b). X* is subdivided into: the "reactive halogen" XO$_x$, grouping X + XO*, and the reservoir species X$_2$, HOX, XNO$_2$, XNO$_3$ and which have a relatively longer lifetime. These last species are formed mostly at night and serve as reservoirs of reactive halogens far from its primary sources.

In the following subsections, we present, for each halogen, the species considered in each reservoir, their respective tropospheric mass burden and the main fluxes between these reservoirs. The purpose is to evaluate the representation of halogenated species of interest for ozone and to discuss the limitations of the model in the computation of the budget of each halogen family.

### 3.2.1 Iodine

Iodine chemistry is the most impactful with regard to the ozone loss. The simplified iodine cycle is shown schematically in Fig. 4. The burden of inorganic iodine excluding IBr / ICl and iodine in aerosols is ~ 22.0 Gg in the troposphere in LMDZ-INCA, comparable to the value of ~ 27.9 Gg reported by Sherwen et al. (2016b, Figure 7). HOI and OIO species, whose photolysis result in atomic iodine production, are the main intermediates for direct ozone loss. HOI is the most abundant iodine species and its wet deposition controls the total atmospheric iodine burden. Its tropospheric burden of 12.8 Gg I in INCA is very close to the 13 Gg simulated by Sherwen et al., 2016a. The tropospheric burden of OIO in INCA, 0.43 Gg I, is comparable to the 0.55 Gg I also reported in Sherwen et al., 2016a. Overall, the global levels of HOI and OIO, simulated by INCA are similar to their simulation by other models described in the literature.

The tropospheric burdens of HI, INO$_2$, and I$_2$ sum up to 1 Gg I, lower than the burdens reported for GEOS-Chem and totaling 2.48 Gg I  (Sherwen et al., 2016a). The global burden of I$_2$ in INCA is lower than that in GEOS-Chem (Sherwen et al. 2016a) because in Sherwen et al. (2016a), heterogeneous reactions transform INO$_3$, INO$_2$ and HOI into I$_2$, whereas in INCA, these

reactions produce IBr and ICl, similarly to the heterogeneous reactions adopted by Sherwen et al., (2016b). Other quantifications in the literature are lacking to explain the differences in $INO_2$ and HI, however, Saiz-Lopez et al. (2014) indicate that minor iodine species ($I_{min}$ = HI + OIO + $INO_2$ + INO) represent less than 5% of the mass in tropospheric inorganic iodine (2.3% in our case). In INCA, the tropospheric $I_y$ mean is 1.3 pptv, whereas previous model studies found a range of ~ 0.5 to 1 pptv (Saiz-Lopez et al., 2014; Sherwen et al., 2016a, 2016b). A large set of observations has been assembled by Sherwen et al. (2016a) for IO. Measurements of IO during oceanic campaigns show a strong diurnal variation (IO being present only during day time) with climatological concentrations (local observations averaged over 20° bins) comprised between 0.3 and 1 pptv. These IO concentrations show a weak latitudinal variability and a rapid decrease with altitude to reach values of about 0.2 pptv from 2km to the tropopause (observations gathered and reported by Sherwen et al. 2016a). Surface values simulated by INCA agree within a factor of two with these observations. Over oceans, the annual mean surface concentrations of IO is approximately 0.2 pptv with concentrations comprised between 0. and 2. pptv. This is similar to the range simulated by Sherwen et al. (2016a) (0.25 - 1 pptv) and to the observations reported by Prados-Roman et al. (2015). In the tropics (20° S – 20° N), levels of surface IO in INCA (0.35 pptv) are in agreement with model results reported for CAM-Chem (~0.4 ppt) (Saiz-Lopez et al., 2014). In the free troposphere, IO levels slightly increase with altitude to reach diurnal values of ~0.5 pptv at 10km in INCA. This overestimation of IO in the upper troposphere is common to all current models simulation iodine and remain unclear (Sherwen et al. 2016a, Badia et al. 2019).

The $IO_x$ family influences the oxidative capacity of the troposphere through the catalytic cycles of ozone depletion (Brasseur, 2005), modifies the $HO_x$ ($HO_2$ / OH) and $NO_x$ ($NO_2$ / NO) ratios (Bloss et al., 2010), produces photolabile higher order iodine oxides ($I_xO_y$) (Gómez Martín et al., 2020; Lewis et al., 2020) and generates different forms of inorganic iodine (Saiz-Lopez et al., 2012a). For that reason, a correct representation of the branching ratios and the generation and loss of $IO_x$ is crucial. Globally, $IO_x$ production in INCA is dominated by inorganic photolysis of HOI (80.5%), OIO (10.2%), $INO_3$ (5.4%) and $I_2O_x$ (3.9%). This partition is comparable to the one reported in Sherwen et al., 2016b who found a 76% HOI photolysis contribution and a 11% OIO photolysis contribution. The main $IO_x$ loss pathway is the production of HOI by IO reaction with $HO_2$ (78.5%), with additional loss pathways by self-reaction, reaction with $NO_x$ and BrO contributing 11.3%, 5.7% and 4.5%, respectively – numbers that are very close to the branching ratios reported in Sherwen et al., 2016b (77%, 10%, 7.7% and 4.6% respectively).

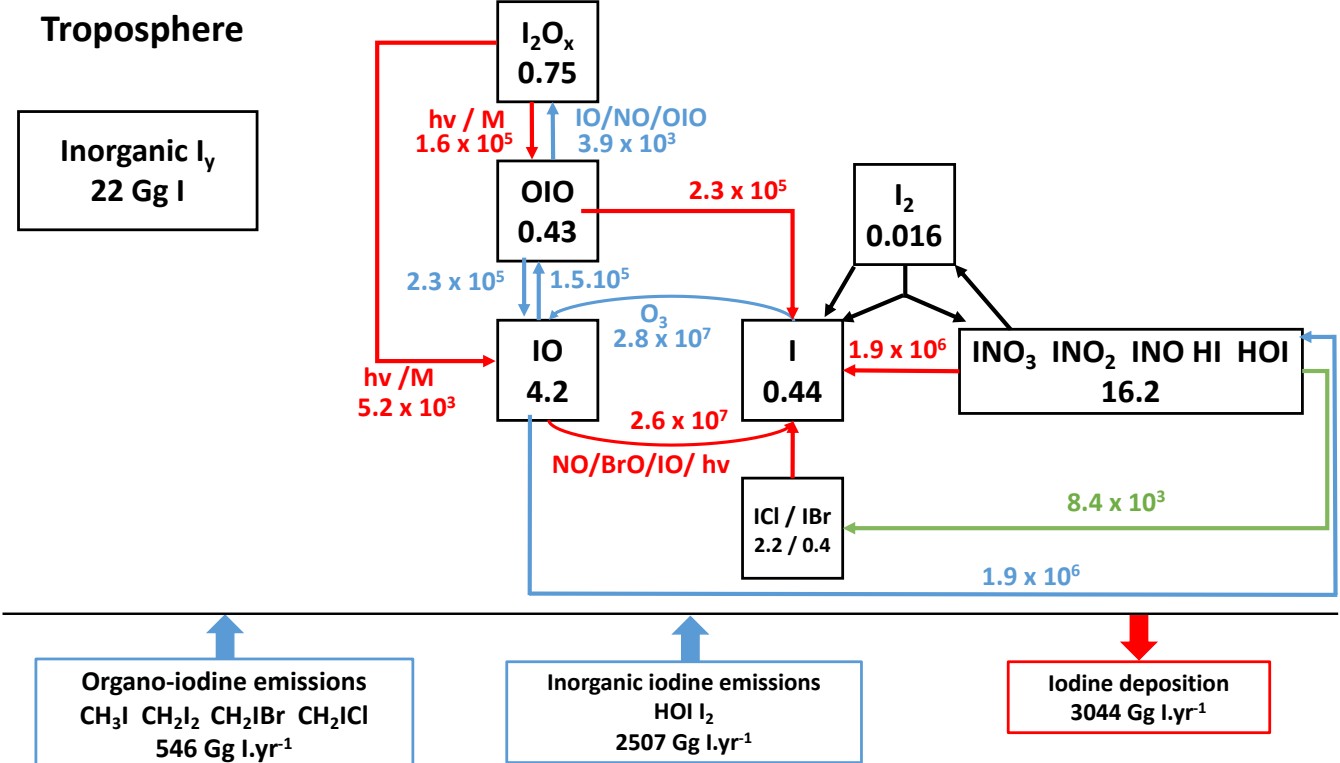

**Figure 4: Iodine budget in the present-day reference simulation with simplified cycle of tropospheric $I_y$ species. Heterogeneous reactions are shown in green. Photolysis reactions are in red. The numbers in the boxes represent the mass balance of the species families in Gg I and the numbers near the arrows represent the fluxes through chemical reactions in Gg I yr$^{-1}$.**

### 3.2.2 Bromine

The tropospheric bromine budget and simplified recycling cycle are illustrated in Fig. 5. The tropospheric inorganic bromine burden is ~ 27 Gg in LMDZ-INCA, close to the 28 Gg reported for the most recent version of GEOS-Chem (Wang et al., 2021). On an annual average, INCA simulates daytime BrO values in the free troposphere of ~0.3 pptv within the tropics, close to the results in CAM-Chem (~0.2 pptv) (Fernandez et al., 2014a). At the tropical tropopause layer, the modeled tropical BrO concentrations in INCA are ~3 pptv, close to the observed concentrations of 2.0±1.5 pptv and of 3.2±1.6 pptv at two different altitudes performed by Dorf et al. 2008 and reported by Fernandez et al., 2014a for the evaluation of CAM-Chem. Overall, the global tropospheric mean of BrO simulated by INCA is 0.6 pptv (in agreement with model results from Parrella et al. (2012)) but is in the lower end of the 0.5 to 2 pptv range inferred from observations (Dorf et al., 2008; Saiz-Lopez and von Glasow, 2012, Stone et al., 2012).

In this first implementation of halogen chemistry in a version of LMDZ-INCA, the focus is on gaseous chemistry. Aqueous chemistry in droplets or on ice crystals in clouds is not accounted for in the first version of the scheme because it is complex and uncertain, and, more importantly, might not be a dominant mechanism for global ozone loss. Nonetheless, the scheme includes recycling through heterogeneous chemistry on sea salt and sulfate aerosols. A study by Badia et al. (2019) using WRF-Chem shows that heterogeneous halogen chemistry (on all types of aerosols) allows a better vertical $Br_y$ partitioning, but contributes little (~6%) to ozone loss compared to the gas-phase halogen chemistry (25%) on a global scale. Our choice to include, in this first step, reactions on sea salt and sulfate aerosols only, is suited to reproduce bromine-mediated ozone loss in the marine boundary layer but probably tends to slightly underestimate the halogen-mediated ozone loss in the global troposphere.

Due to this lack of in-cloud halogen chemistry in cloud droplets and ice crystals, whose importance increase with altitude, the bromine chemistry in INCA is less dynamic and the recycling of bromine is less efficient in LMDZ-INCA compared to those reported for some versions of GEOS-Chem (Parrella et al., 2012; Schmidt et al., 2016; Sherwen et al., 2016b; Zhu et al., 2019). Note, however, that the recently updated HOBr heterogeneous chemistry implemented in the most recent GEOS-Chem version

(Wang et al., 2021) results in much slower recycling of HOBr in clouds and aqueous aerosols than previously calculated. This results in fluxes from heterogeneous reactions (specifically $HOBr + HBr \rightarrow Br_2 + H_2O$ and $HOBr + HCl \rightarrow BrCl + H_2O$) that sum up to 32 500 Gg Br.yr$^{-1}$ and which is smaller for example than our estimates of 86 000 Gg Br.yr$^{-1}$ but remain in the same order of magnitude. Finally, our branching ratios are found to be consistent with those in Schmidt et al., (2016) and Wang et al., (2021). This leads us to conclude that, although LMDZ-INCA does not integrate the same degree of details in tropospheric halogen heterogeneous chemistry compared to its peers, global fluxes in the bromine cycle of INCA appears to be more comparable in terms of bromine cycle with to those in the newest version of GEOS-Chem (see the detailed bromine budget and cycle in Fig. S1) and CAM-Chem (Iglesias-Suarez et al., 2020 ), than fluxes in the older versions of GEOS-Chem (Parrella et al., 2012; Schmidt et al., 2016; Sherwen et al., 2016b; Zhu et al., 2019). All these estimates remain in the same order of magnitude but the range of fluxes illustrates the level of uncertainties on the strength of the halogen heterogeneous recycling.

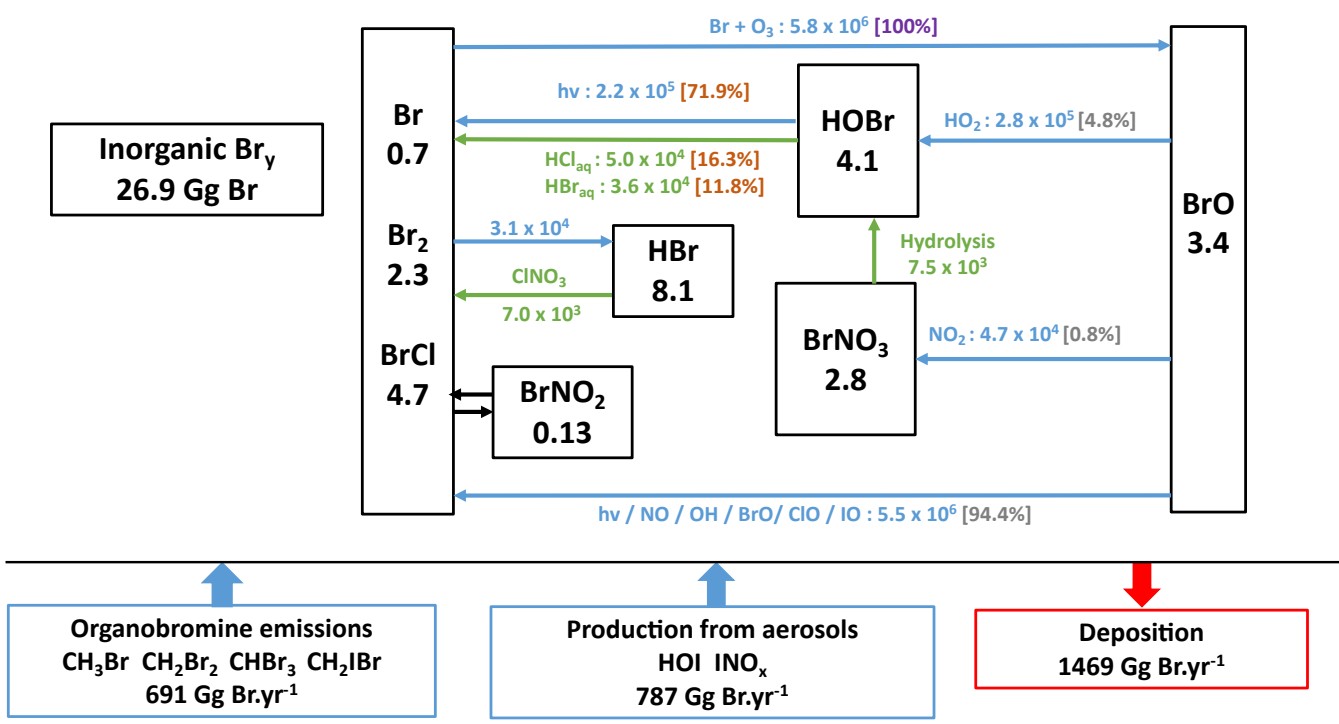

**Figure 5: Bromine budget in the present-day reference simulation with a simplified representation of the cycle of tropospheric Br$_y$ species. Fluxes through the reactions, indicated next to the arrows, are in Gg Br.yr$^{-1}$. Heterogeneous reactions are represented in green. Percentages in brackets represent the branching ratios for the Br chemical sink (in violet), for the BrO chemical sinks (in grey) and for HOBr chemical sinks (on the in light maroon). The numbers in the boxes represent the burden of the species or families in Gg Br and the numbers near the arrows represent the fluxes through chemical reactions in Gg Br yr$^{-1}$.**

### 3.2.3 Chlorine

The total tropospheric burden of Cl$_y$ (inorganic chlorine) is 290 Gg of Cl, largely dominated by HCl. The total load of HCl is smaller than the one reported for GEOS-Chem (Wang et al., 2019) essentially because of the difference in the representation of the emissions. However, the total burden of reservoir species has a similar magnitude (10.4 Gg Cl). The halogen chemistry in INCA leads to a 25% higher quantity of available reactive chlorine compounds: Cl and ClO * (ClO* being ClO+ClOO+OClO+2 Cl$_2$O$_2$).

Atomic chlorine has two main chemical sinks. The first and most important in term of reaction flux, is the transformation of atomic Cl into ClO*. This conversion takes place mainly by reaction with ozone (99.5%), while the conversion of ClO$_x$ to Cl

is dominated by reaction with NO (93.9%). The second path is represented by the chemical loss of reactive chlorine Cl to inactive chlorine HCl (Fig. 6), which is due to the reactions of Cl with methane and several NMVOCs. The conversion of Cl to ClO* is in strong competition with the second route. By dividing the fluxes of the conversion Cl to ClO* by the flux of conversion of Cl to HCl, we find that the chain length is short and equal to 1.3 ($4.8 \times 10^4$ / $3.6 \times 10^4$). This is close to the chain length of 1.6 calculated by Wang et al. (2019) and shows that the branching ratios are in the same order of magnitude in both models, despite differences in NMVOC and $CH_4$ concentrations between the two models. Overall, we find that the partition between Cl and ClO* is correct in the model as well as the balance between radical and non-radical species although the latter is a little more shifted towards reactive species in INCA, compared to GEOS-Chem and CAM-Chem.

The average of climatological observations (mean over several months and representative of large areas) over 6 oceanic and coastal remote locations, shows a mean surface HCl mixing ratio of 196±114 pptv whereas INCA simulates an average of 193±98 ppt, compared to the 246±90 pptv in GEOS-Chem (Wang et al., 2019) (see Fig. S2). However, for continental regions, LMDZ-INCA underestimates the HCl concentrations compared to observations, mainly because it does not include industrial emissions. $ClNO_2$ over continental areas does not exceed a few tens of ppt in INCA whereas observations at several continental locations, as gathered by Wang et al. 2019 (their Table 5), show concentrations of a few hundreds to thousands of ppts over such locations for hourly maxima. This difference can arise from the coarse resolution of the model which does not resolve the spatial heterogeneity of these local observations (Sherwen et al., 2016b) and the absence of $ClNO_2$ production in sulphate-nitrate-ammonium aerosols (Wang et al., 2019). The chlorine Cl* measurements to which we compare LMDZ-INCA results represent the sum of $Cl_2$, HOCl, $ClNO_2$, $ClNO_3$ and other minor components as explained in (Pszenny, 1993). LMDZ-INCA Cl* values are of the same order of magnitude as measurements (see Table S3).

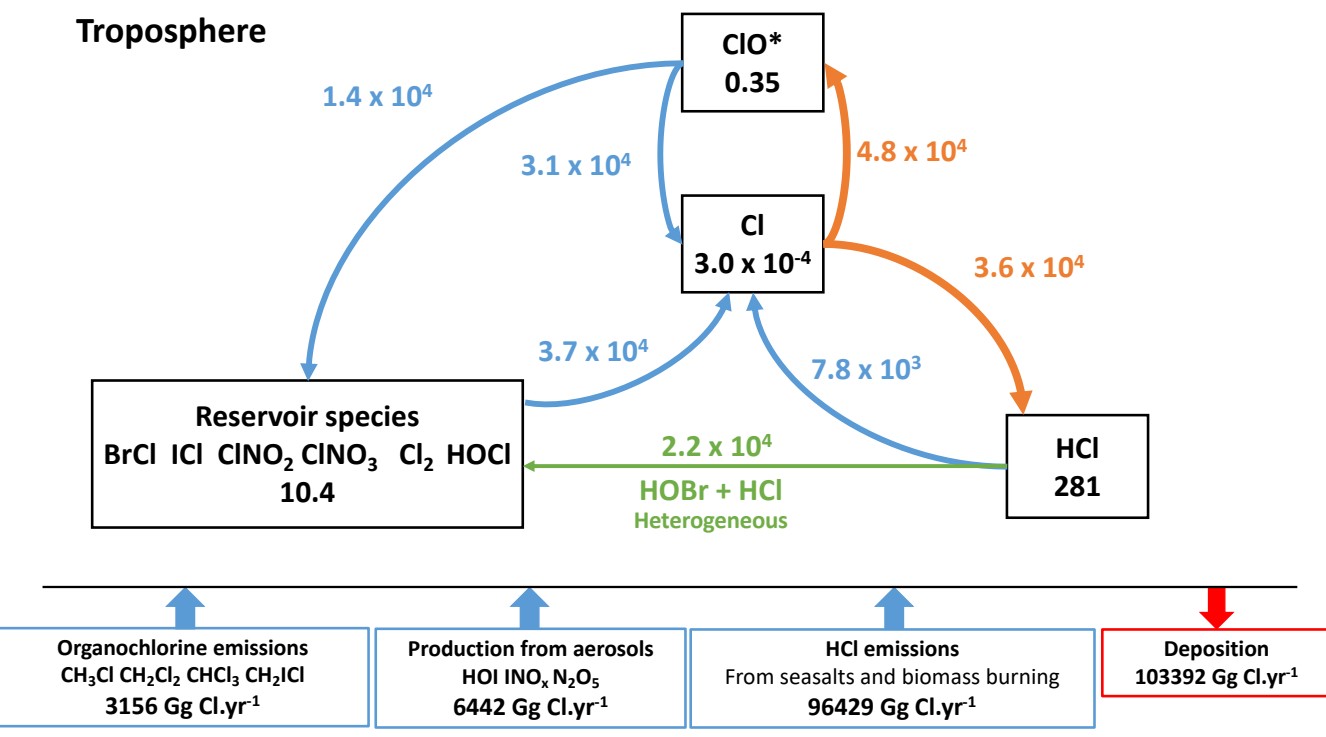

**Figure 6: Simplified representation of the cycle of tropospheric $Cl_y$ species in the reference simulation. Fluxes through the reactions are in Gg Cl.yr$^{-1}$. Heterogeneous reactions are represented in green. The main loss pathways for atomic chlorine are in orange. The numbers in the boxes represent the mass balances of the species families in Gg Cl.**

**4. Perturbation of the ozone sensitivity to its precursors by halogen chemistry**

In this section, the evolution of the model-calculated ozone sensitivity from the pre-industrial period to the present-day is assessed, with and without halogen chemistry, and compared with results from other models. Then the impact of halogen chemistry on the sensitivity of ozone, OH radical and $CH_4$ concentrations to the emissions of anthropogenic ozone precursors is discussed.

**4.1 Pre-industrial to present-day changes**

With and without halogen chemistry, the model simulates significantly lower ozone concentrations in the pre-industrial compared to present due to the decrease in $NO_x$ concentrations. Taking into account halogen chemistry reduces the average surface ozone concentrations for pre-industrial conditions by 5.0 ppbv (38%), with largest reductions over the oceans (5.5 ppbv, or 42.4%).

Globally, halogens reduce the tropospheric ozone burden by 67.1 Tg in present-day and 48.2 Tg in pre-industrial conditions (Table 7). Between PI and PD, ozone increases by 98.1 Tg (113 Tg in Sherwen et al., 2017) without halogen chemistry and by 79.2 Tg (90 Tg in Sherwen et al., 2017) with halogen chemistry. Both in PD and PI, halogens are responsible for about 23-28% of the ozone destruction. Tropospheric ozone lifetimes drop from 21.1 to 17.9 days in the present with the inclusion of halogens and from 25.6 to 21.2 days in the pre-industrial. The sensitivity of the burden of ground-level ozone to pre-industrial – present changes is ~ 20% lower when the chemistry of the halogens is considered. The changes observed in LMDZ-INCA and schematized in Fig. 7 are in agreement with the changes in Sherwen et al. (2017).

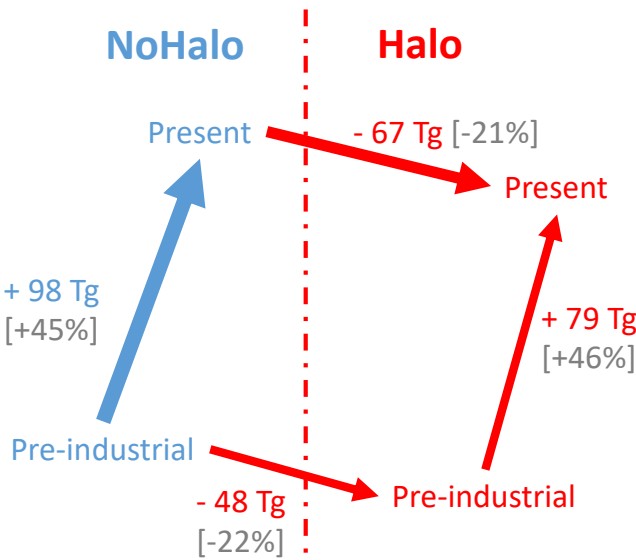

**Figure 7: Schematic representation of the evolution of ozone mass budget changes between PI and PD, following the inclusion of halogen chemistry in LMDZ-INCA.**

## 4.2 Sensitivity runs

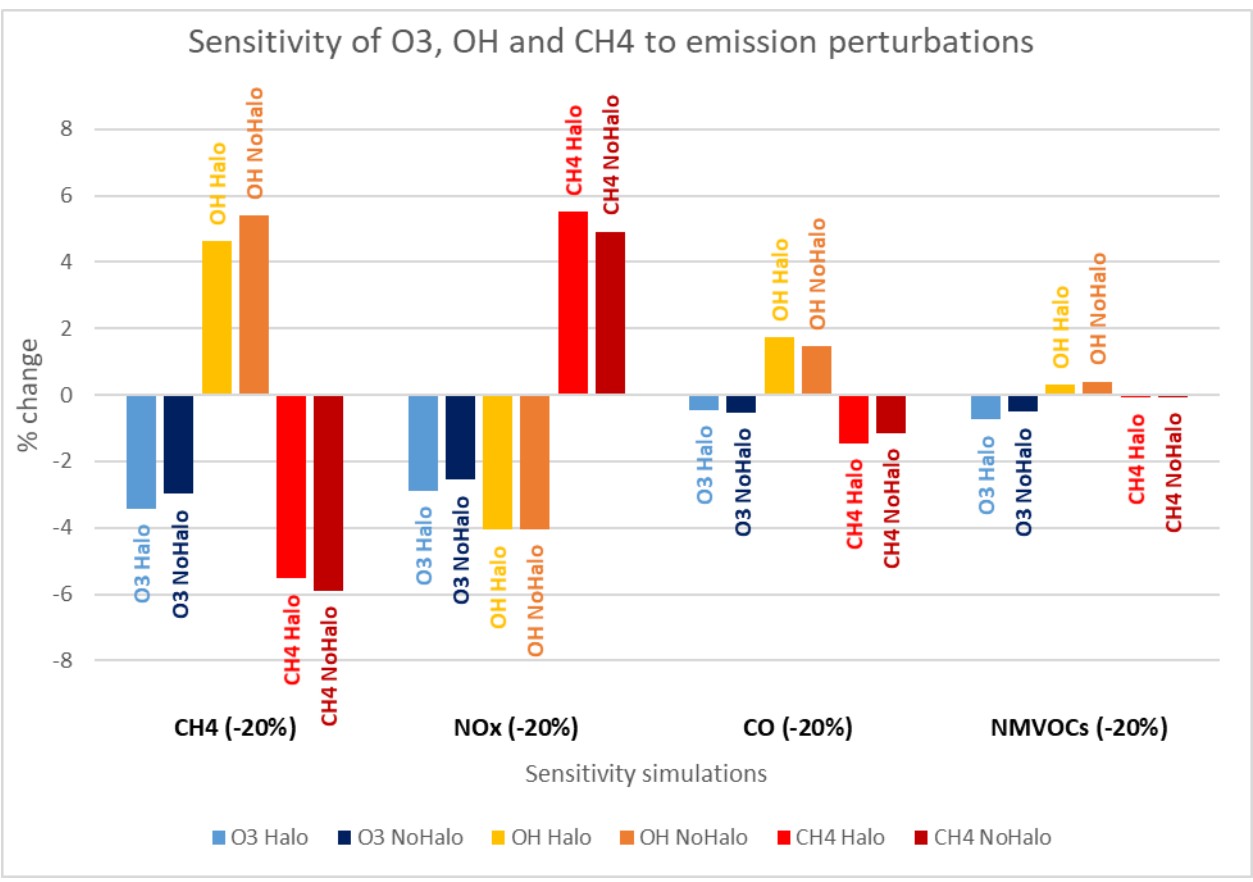

**Figure 8: Tropospheric relative changes (%) in ozone mass budget, OH concentration and methane lifetime for perturbation of NO$_x$, CO, NMVOCs emissions and CH$_4$ concentrations with and without halogen chemistry.**

In order to study the effect of halogen chemistry on the sensitivity of the tropospheric oxidative system, we explore, in Fig. 8, the relative changes (%) of ozone and OH when present-day emissions of photooxidant precursors (NOx, CO and NMVOCs and of CH$_4$ concentrations are perturbed by a 20% reduction, as done in the HTAP protocol)). Our results show that the importance of sensitivities of ozone and OH to individual photooxidant precursors emissions increase as follows in terms of importance: NMVOCs, CO, NO$_x$ and CH$_4$ (Fig. 8). It is also clear that the OH radical is more sensitive to perturbations in CH$_4$, NO$_x$ and CO than ozone. Notably, when accounting for halogens, ozone becomes more sensitive to changes in CH$_4$ and NO$_x$, and to a lesser extent to changes in NMVOCs. On the other hand, in the presence of halogen chemistry, the OH radical becomes more resilient in the presence of halogens to CH$_4$, NO$_x$ and NMVOCs perturbations. Usually, solely the CH$_4$ consumption by OH is considered is when the chemical lifetime of CH$_4$ in the troposphere is reported in the literature since it represents the main CH$_4$ sink. It thus provides a useful proxy for global tropospheric oxidizing capacity (Wild et al., 2020). The simulations examined here show that the CH$_4$ lifetime with respect to OH remains a useful proxy for global tropospheric oxidizing capacity for all perturbations. However, in the presence of halogen chemistry, when NO$_x$ emissions are reduced, atomic chlorine, a secondary oxidant of methane, becomes more efficient and increase the sensitivity of CH$_4$ to NO$_x$ emission change. This exercise shows for the first time the influence of halogens on sensitivity of the tropospheric oxidative system to changes in the emissions of photooxidant precursor (NO$_x$, CO and NMVOCs and of CH$_4$).

## 4.3 Impact on methane lifetime

The inclusion of tropospheric halogens affects the atmospheric concentrations of a large number of compounds in the model. A marked decrease in oxidants (O$_3$, OH, HO$_2$, H$_2$O$_2$) is then computed. This leads to an increase in the concentrations of CO (+8.5%) and some NMVOCs, including isoprene (+7.1%). The lifetime of methane with respect to chlorine and the OH radical as well as the share of chlorine in the losses of methane in the literature are shown in Table 8. In the reference simulation, the

tropospheric chemical lifetime of methane due to its consumption by OH is of 7.9 years. With the inclusion of halogen chemistry, the OH concentration decreases, increasing this chemical lifetime of methane to 9.3 years. The chemical lifetime of methane linked to oxidation by Cl is 333 years, less than the 384 years suggested by Hossaini et al. (2016) but closer to

IPCC AR5 estimates. This oxidation by Cl represents 2.7% of the total loss (close to the values reported in the literature and in Table 8). Thus, in LMDZ-INCA, accounting for both OH and the chlorinated radicals consumption, methane oxidation leads to a total chemical lifetime of $CH_4$ of 9.0 years.

**Table 8: Methane lifetime with respect to chlorine and OH radical as well as the share of chlorine in methane losses in the literature.**

| | Lifetime relative to | | Total chemical lifetime (years) | Method | Share of chlorine in methane loss (%) |
|---|---|---|---|---|---|
| | Chlorine (years) | OH radical (years) | | | |
| *AR5 IPCC (2013)* | $200 \pm 100$ | $11.2 \pm 1.3$ | $9.25 \pm 0.6$ | Multi evidence assessment | - |
| *Hossaini et al. (2016)* | 384.4 | 10 | | TOMCAT model | 2.5 |
| *Sherwen et al. (2016b)* | - | 8.28 | 8.16 | GEOS-Chem model | 2.0 |
| *Li et al. (2022)* | | 10.6-10.8 | | CAM-Chem model | |
| *Szopa et al. (2021)* | | $9.7 \pm 1.1$ | $9.1 \pm 0.9$ | Multi evidence assessment | |
| ***This work*** | - | **7.9** | **7.9** | **LMDZ-INCA model (NoHalo)** | - |
| | **332.5** | **9.28** | **9.0** | **LMDZ-INCA model (Halo)** | **2.7** |

## 5 Conclusions

Global chemistry climate models allow to simulate the evolution of the atmospheric composition and are used for future projections in the framework of international model intercomparisons such as AerChemMIP. Most of these models do not account for halogen chemistry. We have implemented a simplified representation of the halogenated chemistry in the three-dimensional climate-chemistry model LMDZ-INCA to account for its effects on the tropospheric chemistry, notably ozone. First, the halogenated species, their sources and sinks, their reactions in the gas and heterogeneous phase and their

physicochemical characteristics have been integrated into INCA. In order to simplify the integration of this chemistry, a chemical scheme already tested and evaluated in the CAM-Chem model has been used as a basis for our developments. The representation of this chemistry has been shown to be sufficient in a first approach to simulate properly the impact of halogens on the photooxidizing system in the troposphere and particularly in the atmospheric boundary layer. The response of the tropospheric chemical system $O_x$, $HO_x$, $NO_x$, $CH_4$ and NMVOCs to incorporation of halogen chemistry has been quantified.

On a global tropospheric scale, this chemistry induces decreases of the ozone burden by 22%, of OH by 8% and of $NO_x$ by 33%. An increase in CO and NMVOC concentrations are simulated and the lifetime of methane increases by one year. These changes are consistent with those reported in the literature. Comparisons between simulations considering preindustrial and present-day emissions show that the sensitivity of the ozone burden to changes in emissions over the preindustrial era is $\sim 20\%$ lower in the presence of halogen chemistry (mainly because of the chemistry of iodine).

Sensitivity tests, consisting on reducing in turn the emissions of ozone precursors and methane concentrations, show for the first time that ozone becomes more sensitive to $NO_x$, CO and NMVOC perturbations in the presence of halogen chemistry. Although still rarely represented in chemistry climate models, our model results confirm that the chemistry of tropospheric halogens (Cl, Br, I) plays a significant role in the global loss of tropospheric ozone. Since halogen chemistry affects substantially the ozone burden both in preindustrial and present-day chemical conditions and the ozone sensitivity to changes

in anthropogenic emissions, international exercises examining the evolution of ozone between preindustrial, present, and future should consider to take into account halogen chemistry.

**Data availability**

The main model codes and the chemical scheme with halogenated compounds as well as the data outputs are publicly accessible through the doi: 10.14768/56b1b070-deae-4357-a2dd-d75d27659f63

**Author contributions**

The implementation of the halogens chemical scheme has been done by CCaram, SS and AC. The design of the numerical simulations was conducted by CCaram, SS, AS-L and CCuevas. CCaram and SS prepared the Figures. CCaram, SS, SB, AS-L and CCuevas analyzed the results and drafted the paper.

**Competing interests**

The authors declare that they have no conflict of interest.

**Acknowledgements**

The authors are thankful to the NCAR Atmospheric Chemistry Division (ACD) for the distribution of the NCAR/ACD TUV:
Tropospheric Ultraviolet & Visible Radiation Model (http://cprm.acom.ucar.edu/Models/TUV/Interactive_TUV/) and the availability of their quicktool.

**Financial support**

This research has been supported by the Agence Nationale de la Recherche (PALEOx project, grant no. ANR-16-CE31-0010).
This work was granted access to the HPC resources of TGCC under the allocation A0090102212 made by GENCI (Grand Equipement National de Calcul Intensif). This study received funding from the European Research Council Executive Agency under the European Union's Horizon 2020 Research and Innovation Programme (Project ERC-2016- COG 726349 CLIMAHAL).

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
