# Peer review of "Sensitivity of tropospheric ozone to halogen chemistry in the chemistry-climate model LMDZ-INCA vNMHC"

_Geoscientific Model Development, 2021_

## Author Comment (AC1)

**In Black the reviewer comments and in blue our responses**

In this manuscript, Caram and colleagues present the reactive halogen chemistry in the chemistry-climate model LMDZ-INCA, focusing on the sensitivity of tropospheric ozone. This work emphasizes a long-standing issue in the climate community that the impacts of halogen chemistry on key short-lived climate forcers (such as ozone and methane) have not been fully accounted for in most climate models.

The method section of this manuscript lacks key details of the model (LMDZ-INCA) that are relevant for the scientific goal of this work. For example, acid displacement is a major source of reactive chlorine in the atmosphere. To properly model acid displacement, several key processes need to be properly implemented in the model, such as the emissions, dry/wet depositions, and transport of sea salt, as well as its physical and chemical properties (especially acidity), not to mention the acidic gas(es) to displace HCl (e.g., nitric acid). It remains unclear how these processes are implemented in this model LMDZ-INCA. It does appear that the model uses highly simplified HCl emissions to account for this process, yet no details are given (i.e., spatial/temporal variability of such emissions). Similarly, debromination is a major source of reactive bromine in the troposphere. Yet, how this process is implemented in LMDZ-INCA is unclear. Fig 3 looks like the model only recycles HBr produced from other reactive bromine species (not a net bromine source); if this is indeed the case, essentially the only source of the reactive bromine is the organobromines. Key details like these should be clearly stated.

The paper explains how we implemented the chemistry of halogenated compounds in a global tropospheric chemistry model LMDZ-INCA, an important component of the IPSL Earth System Model (ESM). LMDZ-ICA is often used for multiple long chemistry-climate simulations. We explain the methodology adopted, the capabilities and limitations of using such a model.

We should have stressed in the introduction that our aim was not to implement a very comprehensive description of tropospheric halogen chemistry with all the details or to simulate accurately the temporal and spatial evolution of all the halogenated compounds. The aim was to develop a simple and computationally cheap halogen chemistry scheme which is able to reproduce most of the effects of halogen species on climate-relevant species (e.g. ozone, methane), and then to assess for example whether iodine, bromine and chlorine chemistry can, beyond their already demonstrated impact on tropospheric ozone, also affect the sensitivity of ozone to its precursors, a potentially important considerations in chemistry-climate interactions. Having simplified models with an acceptable computation time makes it possible to carry out a large number of long global chemistry-climate simulations. The challenge is to find the right compromise between the level of realism and details (e.g. number of species and processes) acceptable for the chemistry-climate applications, and the computing efficiency. Our choices of chemistry representations and emissions are guided by results from global tropospheric chemistry models that include very detailed representations of halogen chemistry. Their results, notably the impact on the tropospheric ozone budget, are used as benchmarks in assessing the LMDZ-INCA simulations.

The heterogeneous component is the most complex part of halogen chemistry. It is also by far the most uncertain. In addition, we implemented the chemistry of I, Br and Cl in a version of INCA where aerosols (e.g. number/surface/mass concentration, composition) are not

prognostic variables but prescribed through climatologies. Nonetheless, although the (missing) explicit treatment of some heterogeneous processes (acid sensitivity of sea-salt de-chlorination and heterogeneous reactions on aqueous aerosols and ice-crystals in clouds) can be important, especially to simulate the vertical distributions of halogenated compounds, it does not drive the overall impact of halogen chemistry on global tropospheric ozone in the model according to the comparisons with the more detailed reference chemistry-transport models (GEOS-CHEM and CAM-Chem). The comparisons show that, despite a simplified treatment of the heterogeneous halogen chemistry, our scheme is able to simulate satisfactorily the global levels of halogens and to account for the overall effect of halogen chemistry on tropospheric ozone. It represents a major model improvement still missing in almost all the global chemistry-climate modelling studies. As pointed out by the reviewer, it is "a long-standing issue in the climate community that the impacts of halogen chemistry on key short-lived climate forcers (such as ozone and methane) have not been fully accounted for in most climate models."

We recognize that the language was sometimes vague and some formulations were clumsy. We sincerely apologize and thank the reviewer for his/her recommendation. We did a thorough work on the manuscript in order to clarify the intent and better organize it to make the flow more logical. Below we give more details on how we respond more specifically to these general and further specific comments.

This work does take other more established models as benchmarks, especially GEOS-Chem and CAM-Chem. The halogen chemistry implemented in this model (LMDZ-INCA) appears to be much simpler than GEOS-Chem or CAM-Chem.

The reviewer is correct. However, note that only the heterogeneous component of our halogen chemistry scheme is much simpler than in GEOS-Chem or CAM-Chem. It also is the most complex and uncertain part of this chemistry. The level of complexity of our heterogeneous representations was also limited by the fact that aerosols were prescribed instead of being calculated in our version of LMDZ-INCA. As a consequence, some heterogeneous processes could not be represented in a very detailed and interactive way, especially compared to GEOS-Chem or CAM-Chem in which the descriptions of halogenated tropospheric chemistry result from years of development. However, we show that our model is sufficient for some chemistry-climate applications, in particular investigating the impact of halogen chemistry on tropospheric ozone and its sensitivity to its main anthropogenic drivers. This is a novel result of our study.

The measurement-model (LMDZ-INCA) comparison is not adequately transparent and contains almost no specifics. A few major issues: the measurements and observations are not presented in consistent manners; the comparisons completely ignore the spatial and temporal variabilities revealed in the observational datasets. The level of detail presented in these comparisons does not adequately justify the performance of this model.

LMDz-INCA is a global model and, to keep computing costs reasonable in this development phase, it is used at a coarse resolution. At this stage, we assess the ability of the model to reproduce climatological datasets (i.e. averaged over time and location) which is the most appropriate scale for a first level of evaluation of a global chemistry-climate model. As explained previously, although we have not performed a comprehensive comparison with observations, the evaluation of our model performances relies largely on comparisons to

more complete and established models, CAM-Chem and GEOS-Chem, which have already been rigorously evaluated in a detailed manner against a wide variety of observations (as pointed out by the reviewer). The correct simulation of the global halogen species loading and of its contribution to the ozone budget, compared to CAM-Chem and GEOS-Chem, illustrates the relevance of our scheme, as implemented in LMDz-INCA.

Please see the next section for details. It is my opinion that this manuscript is not well written either, with numerous vague/confusing statements, which I will elaborate later. In light of these, it is unclear in the current manuscript what new scientific insights this study offers, compared to other models such as GEOS-Chem and CAM-Chem, which are more comprehensive and have been rigorously evaluated against a wide variety of observations.

Our aim is not to develop a scheme matching or more sophisticated than GEOS-Chem and CAM-Chem or carry out specific chemistry process studies with detailed comparisons against observations. The focus is on a simplified model with reasonable computational costs for chemistry-climate applications, in particular those involving the impacts of halogen species on global tropospheric ozone and oxidizing capacity. This also offers new scientific insights from the point of view of chemistry-climate interactions involving halogens.

This study introduces another model, an important component the IPSL ESM, which can be used for certain applications if not for all. Even with some limitations, the new scheme clearly helps to improve the ozone representation in our model, which is an important milestone in the model development. GMD is well suited for this type of work because it is 'dedicated to the publication and public discussion of the description, development, and evaluation of numerical models of the Earth system and its components'. We are convinced of the importance of this newly implemented chemistry scheme in LMDZ-INCA, to better assess the change in tropospheric ozone since the preindustrial era or in the future.

The necessity to have model ensemble is now well established and it requires the development of a diversity of models. Having already the halogenated chemistry in three global models cannot be a reason not to attempt to implement it in other models. There is not much scientific insight to gain if we all use the same models, perform the same studies and hence end up with the same results. Finally, even if it is just an illustration of the possible model applications, our results on the change in tropospheric ozone sensitivity to anthropogenic precursors produced by halogens are valuable and bring some new scientific insight on potential chemistry-climate interactions involving halogens (section 4.1).

The way that the pre-industrial simulations are configured is based on key assumptions that are unverified/unjustified. For instance, this study essentially assumes several key processes of the Earth system (e.g., lightning, biomass burning, biogeochemistry in the ocean) remain the same between the pre-industrial period and the present-day. I am not fully convinced.

The main purpose of this part is to illustrate how the consideration of halogenated chemistry can perturb the model-calculated response of tropospheric ozone to changes since preindustrial times. First, regarding the ozone burden evolution, the main driver for atmospheric chemistry between preindustrial and present-day is definitely the emissions from anthropogenic sources. Second, emissions from natural sources have also changed but are not the primary driver, and the uncertainties around such changes are large. There is no

general consensus on the magnitude and sometimes even the sign on some of these changes.

One specific example is again acid displacement largely driven by HNO3, which, may be different between pre-industrial periods and present-day. These key assumptions need to be justified.

The aim was not to reproduce as faithfully as possible the distribution of the Cl-containing compounds. As the chlorine chemistry is less reactive than the bromine and iodine chemistries, it was not considered as a priority in this development focused on the impact of halogens on ozone. For chemistry-climate applications, chlorine is essentially significant as an additional sink for methane (which was estimated to 2 to 3% in recent literature not accounting for acid displacement and 1.0 % of the total methane chemical loss when considering the pH dependant chlorine chemistry (Wang et al. 2019)) but chlorine chemistry only plays a minor role in the halogen-driven ozone loss. The main reason for implementing some chlorine chemistry was to represent correctly the bromine recycling due to chlorine through the heterogeneous reactions involving HCl and BrCl.

The chemistry involving acid displacements leads essentially to a source of HCl from sea salt aerosols. This source is accounted for but its dependency to pH is not explicitly computed in an aqueous chemistry module. However, our total chlorine emissions (96Tg Cl.yr-1), which corresponds to the flux in TOMCAT by Hossaini et al. (2015), is realistic and and hence the global effect of chlorine on ozone and oxidation capacity is reasonably well reproduced in our model.

To consider acid displacement of HCl interactively in a realistic way, the LMDZ-INCA model should also account for aqueous-phase chemistry to consider reactions with Cl-. Aqueous-phase chemistry is beyond the scope of this first development phase, especially with the aim of minimising complexity and computational costs.

It is worth pointing out that this particular aqueous chemistry was added in GEOS-Chem only very recently (Wang et al., 2019) and thus our work is more comparable to the level of model descriptions reached in GEOS-Chem in Sherwen et al., 2016b and Sherwen et al., 2017. However, when we discuss the results from LMDz-INCA, we compare them to Wang et al. (2019) whose model version successfully simulates the observed mixing ratios of HCl. INCA is able to reproduce HCl concentrations of a hundred ppt over remote locations and the model-observation discrepancies are of the same magnitude as in GEOS-Chem. As the acid sensitivity is missing in INCA, the model is not able to faithfully capture the spatial variability of the mean HCl mixing ratios across locations, in particular the HCl enhancement at polluted coastal sites and northern midlatitudes. In GEOS Chem (Wang et al. 2019), HCl mixing ratios are highest over the oceans downwind of polluted continents due to the effective acid displacement from sea salt aerosol by HNO3 and H2SO4.

Therefore, I do not recommend this manuscript for publication in Geosci. Model Dev. Please find my specific and technical comments below:

Line 26: Indeed the earliest study of the reactive halogen chemistry in the troposphere was perhaps a decade later than that for the stratospheric halogen chemistry, but since then there have been quite a few large-scale, comprehensive field experiments focusing on the

tropospheric halogen chemistry. Therefore I do not think "…halogens in the tropospheric chemistry had been investigated but to a lesser extend…"

The sentence was qualitative and not intended to minimize the efforts which have been done in particular in term of field experiments. We have removed "but to a lesser extent than for stratospheric chemistry".

Line 40: reactive halogen species also greatly affect the chemical transformation of atmospheric mercury.

yes, we agree that's why it is listed in the sentence

Line 51: halogen chemistry is also active in polluted atmosphere, e.g., Thornton et al 2010.

This paragraph aims to make the point of the importance of halogen chemistry in representing the overall chemistry of the pristine atmosphere.

Line 58-59: please define HTAP, ACCMIP, and AerChemMIP. This is a great point. Please also elaborate how these model intercomparison projects have been used to evaluate the effects of international policies.

This point has been elaborated.

Line 87: consider deleting non-methane hydrocarbons (NMHC) since this category overlaps with volatile organic compounds.

Thank you, it has been done.

Line 89: define ORCHIDEE.

Thank you, it has been done.

Line 101: heterogeneous chemistry on sea salt: is this simply halogen recycling (i.e. not a net source) or debromination (net source)? Please clarify.

We have replaced "heterogeneous chemistry" by "Recycling through heterogeneous chemistry" to clarify. Note that the chemical scheme is now entirely provided in the manuscript itself (table 5).

Line 105: "ice chemistry" is vague. What type of ice? What kind of chemistry on ice? What microphysics option/package is used? Please clarify. Also, I do not quite follow how discarding ice chemistry can *ensure* bromine cycling in the MBL.

We apologize for the lack of clarity here. We mean reactions on ice crystals in clouds. We have rephrased the sentence here to clarify and removed the sentence about ensuring cycling in MBL which was confusing in this context.

We have also clarified in part 2.1.1. that the version of INCA used here does not compute aerosols interactively, climatologies are specified.

Line 106-107: Does this mean that the model cannot reproduce bromine-mediated ozone loss if the so-called "ice chemistry" is included? This may be important information for the community.

As assessed by Badia et al. (2019) the loss of Ox due to the heterogeneous reactions of halogens (mainly bromine) is 6% of the total loss for the whole troposphere. In the marine boundary layer, these reactions account for about 11 % of ozone losses (see figure 15 in Badia et al., 2019 comparing the WRF-NOHET and WRF-DEBROM runs and text in their part 5.4). In-cloud chemistry (on clouds droplets, ice crystals) is the most important in the upper troposphere (Badia et al. 2019), but the loss of Ox due to the heterogeneous reactions involving halogenated compounds falls to less than 5%. Overall, most of the Ox losses by halogens take place in the lower layers of the troposphere where reactions on sea salts dominate and where the in-cloud chemistry has a minor effect.

Line 135: rephrase.

It has been rephrased.

Line 181-196: in this section (chlorine), only a few ratios are roughly compared to a single model (GEOS-Chem) on the global scale (I guess the numbers are global means). Indeed the GEOS-Chem study cited here (Wang et al., 2019) is fairly comprehensive and probably state-of-the-art, but the level of detail presented in this manuscript does not adequately justify the performance of this model (LMDZ-INCA). Note that the GOES-Chem study cited here (Wang et al., 2019) have been evaluated using a wide variety of observations (ground-based: particulate chloride, gaseous HCl, HNO3, ClNO2; airborne: HCl, ClNO2, HOCl, Cl2). I fail to understand why the rich reactive chlorine dataset used for model evaluation in Wang et al. (or even a subset of it) is not used to evaluate this model (LMDZ-INCA).

See above. The aim is not to reproduce Wang et al., 2019 study. GMD would not be the most suited journal for such a process study.  LMDz-INCA is used here in a coarse resolution 3.75°x1.9° and is run over 1 year. Evaluations of global models against observations make sense only if these observations are gathered/averaged spatially and temporally to be representative of climatological compositions, on scales relevant to global chemistry-climate modelling. For the sake of simplicity, we compare the results from INCA to observational datasets already gathered in the literature by modellers who have the same purpose and we also compare the results to the most advanced versions of the models considering tropospheric halogenated chemistry.  Again, chlorine chemistry is not central to the paper which is focused on ozone loss, mainly dominated by iodine and bromine.

Line 186: "atomic chlorine evolves… " this is confusing and inaccurate from the chemistry perspective. Consider rephrasing to something like "atomic chlorine has two main sinks/reaction pathways/…"

OK, it has been done.

Line 202: What is considered as "efficient"? How is this efficiency quantified in this model and in the cited GEOS-Chem studies? Note that the bromine schemes are very different in these GOES-Chem studies (Parella et al., 2012, Schmidt et al., 2016; Zhu et al., 2019).

The efficiency is assessed by quantifying the mass fluxes through the reactions. Fast production and loss of bromine radicals lead to large fluxes and thus an efficient recycling. A more detailed (but still schematic) representation of the bromine reaction fluxes in LMDz-INCA and in GEOS-chem (Schmidt et al. 2016) has been added to the supplementary material (Figure S1). Reaction fluxes are in a relatively good agreement despite differences in the representation of the chemistry.

Line 210: This entire paragraph is problematic: 1) "Since our goal is to simulate the impact on global and ground-level ozone, rather than simulating a state-of-the-art partition within inorganic bromine…" I fail to follow the logic here. What is a point of simulating the impact on global and ground-level ozone if the model does not represent the state-of-the-art understanding of bromine chemistry? Especially, the "impacts on global and ground-level ozone" have been examined using more comprehensive models (like GOES-Chem and CAM-Chem).

We did not state clearly that the aim was to develop a simple and computationally cheap halogen chemistry scheme which is able to reproduce most of the effects of halogen species on climate-relevant species, i.e. ozone, methane via oxidizing capacity, and then explore how halogen chemistry can affect the sensitivity of ozone (and the related oxidizing capacity) to its precursors, a potentially important considerations in chemistry-climate interactions. The challenge is to find the right compromise between the level of realism and details (e.g. species and processes) acceptable for the chemistry-climate applications, and the computing efficiency for the long chemistry-climate simulations. Therefore, we only represent the reactions and species which are dominant for the impact of halogens on global ozone destruction. Some reactions can be of importance in specific environments but only marginally impact global ozone destruction and thus were not prioritized in such development. None of the global models which have been used in recent multi-model projections of atmospheric chemistry for climate takes into account the halogenated chemistry despite the fact they assess ozone budget. The fact that 3 models have developed, over more than ten years, a very comprehensive representation of halogenated chemistry does not mean that other models should not start such implementation even if gradually and thus, in the first stage, less comprehensively. A figure (Figure 3) has been added to show the Ox destruction in the troposphere in INCA. It can be compared with similar figures for GEOS-chem (Sherwen 2016a for scheme with Br and I chemistry and Sherwen 2016b fro Br,I and Cl chemistry). It illustrates that the model is able to reasonably simulate the effect of halogenated chemistry on ozone.

We modified the introduction and flow of the paper to better reflect its intent.

2) "… we make sure that the bromine cycle is chemically active…" This sounds like a very low bar. One can have an active bromine chemical mechanism as long as a few key reactions are included. 3) "…even though reaction fluxes are a factor of 1 to 2 smaller…" please clarify what fluxes the authors are referring to here.

The sentence has been rewritten "The branching ratios are found to be consistent with Schmidt et al., (2016) even though reaction fluxes can differ up to a factor of 2"

Line 222: "the concentrations of HI, INO2, and O2 are lower…" where can the readers find the concentrations of these compounds??

In the sentence "concentrations" has been changed in "burden" and a more detailed figure is now shown.

Line 230: if the authors do not bother to clarify how geographical representation of organic/inorganic iodine sources affect iodine burden, might as well delete this sentence because it serves no purpose as currently written.

The sentence has been removed.

Figure 2-4: these schematic diagrams are so simplified that they are very difficult to follow and may be confusing for readers who are not experts in this area. I'll list a few here: 1) Many important pathways are not shown (e.g., the production of HBr, BrNO2 chemistry). 2) In Fig 3, BrO photolysis and the reactions with other compounds (NO, BrO, ClO) are not separated and some reactions may lead to a net loss of ozone (or odd oxygen) but some don't. 3) In Fig 3, Br (a very short-lived radical), Br2 and BrCl (both stable compounds) are placed in one box, and it's unclear where some of the arrows are pointing at. 4) it is not immediately clear how the bottom center box (HOI, INOx, N2O5) affects chlorine (Fig 2) and bromine budget (Fig 3).

These figures aim to present the budget. Only a few chemical pathways and the main compounds or reservoirs are represented. The chemical scheme for halogenated compounds is now included in the main text as tables 1-5 (instead of supplementary material). Some compounds/fluxes have been added (e.g. BrNO2, HBr production) The legends now make clear that it represents the main terms in the budget rather than the complete tropospheric cycle.

5) It remains absolutely unclear what "physical loss" is and how it works, e.g., is this wet or dry deposition?? What compound(s) exactly?

Physical losses are wet and dry deposition. It has been clarified in the Figures.

6) the number formatting is very confusing. For instance, the Cl burden in Fig 2 is written as "3,0.10^-4". I am aware that comma is also commonly used to separate the integral part of the number from the decimal part (which is perfectly fine), but then the full stop is used in the scientific notation (not even the dot multiplication symbol). All together it looks like two numbers (3 and 0.10^-4). Please follow the commonly accepted number formatting in scientific publications as well as journal guidelines.

Corrected.

7) The percentage values in the square brackets are not defined.

The percentages correspond to the branching ratio for the sinks of HOBr and BrO, it has been clarified in the legend.

8) In Fig 4, not one single reaction rate is given (while Figs 2 and 3 do list a few rates) and what exactly are "gas phase reactions"??

Added to the iodine figure in the manuscript.

Line 247-248: this sentence offers virtually no specifics. Where are these sites located? Where are the ClNO2 comparisons presented?

"LMDZ-INCA underestimates the continental $ClNO_2$ content for many sites" has been replaced by "ClNO2 over continental areas does not exceed a few tens of ppt in INCA whereas observations at several continental locations gathered by Wang et al. 2019 (their Table 5) show concentrations of a few hundreds to thousands ppt over such locations for hourly maximums. "

Table S5: the last four reactions appear to rely on underlying assumptions. Essentially the yields and branching ratios used in these four reactions need to be justified. Also, some of these reactions do not obey mass conservation, e.g., HOI -> 0.85 ICl + 0.15 IBr + HNO3. This reaction makes chlorine, bromine, and nitrogen out of nothing, which might not be complete nonsense under very specific conditions but are certainly dangerous and should be avoided in modern chemical transport models.

This is the way it is considered in GEOS-chem as well. These are not elementary reactions but a representation of dechlorination and debromination of seasalts, and the resulting bromine and chlorine in that specific reaction comes from the chloride and bromide in the sea-salt particle.

Table S9: The way the comparisons are conducted in this table does not make much sense and definitely is not "satisfactory" (Line 251). 1) Is Cl* the sum of total reactive chlorine or some individual compound (Cl2, HOCl, ClNO2, ClNO3)? Either way, such comparison offers practically no scientific insights. I fail to understand why the comparisons are not conducted for each individual compound.

-> We do not make comparisons to individual compounds because the papers we compare the results to report the values in a similar way. We had to adapt to other papers' results. We use for Cl* the same definition as Wang et al., 2019.

2) if the measurements and are shown as 24-hour means, please also show standard deviation or something like that to represent the variability of these observations. 3) I do not understand why monthly means (modeled) are compared to 24-hour means (measured)?

Measurements correspond to average of several 24hours mean measured over 1 or 2 months. These results have been aggregated by Wang et al. 2019, we use them in a similar way.

Line 253: Table 2 is also problematic. 1) the measurements are shown in very different styles: some given in a range, some with a plus/minus (unclear what these are), and some are given a single value. The table caption does not help.

Not applicable, the Table has been removed.

2) the use of "<" is also confusing. It says in the caption that this symbol indicates that mixing ratios are below detection limit, yet in the table the symbol appears to be used differently. For instance, in the first line of the table reads like the measured BrO was below 2.0 ppt.

Not applicable, the Table has been removed.

3) many of the observations contain rich information on the spatial variability (vertical, geographic) which is wasted completely here. Why don't the authors show a few vertical profile plots for the airborne measurements and a few time-series or alike for the ship-based measurements?

The model has a 3.75°x1.75° resolution, it does not make sense to compare to individual observations. Observations can be used if they have a climatological representativity.

4) again it makes no sense to compare the modeled diurnal averages to the measured daily max.

Not applicable, the Table has been removed.

Line 279: averaged across the entire atmosphere or troposphere or what?

It has been clarified in the text

Line 282: why ozone column carries the unit of ppbv?

The formulation was clumsy, it has been modified to avoid confusion.

Line 303-304: the total ozone chemical annual loss decreased by 2.4% in the simulation with halogen chemistry, I assume?

Sorry, the sentence has been clarified.

Line 347: please provide evidence (perhaps a few citations) to support this statement.

For sake of clarity, this paragraph has been removed.

Line 349: what is this "NOx concentration threshold"?

For sake of clarity, this paragraph has been removed.

Line 350: "… NOx from 24-27 pptv to 39-41 pptv" it is completely unclear what these values are.

For sake of clarity, this paragraph has been removed.

---

## Author Comment (AC2)

**In Black the reviewer comments and in blue our responses**

The paper in review for GMD by Caram and co-workers is a first attempt to include tropospheric halogen chemistry within the global chemistry-climate model LMDZ-INCA. This is a very interesting and well intended effort. But, the current version and modeling fall short of the high standard of GMD. I agree with reviewer #1 on all points, but come to a different conclusion that if the modeling and the presentation within the paper are improved, this can eventually be published in GMD with major revisions.

The main problem with the paper at present is that there are no new scientific conclusions that can be drawn from this modeling effort. I suppose that GMD papers can be based mainly on descriptions of new model implementations, but in the case that this is acceptable, then the model description has to be significantly improved in it's clarity and some errors and omissions must be addressed for this to be acceptable.

We apologize for the confusion. We did not state clearly our intent. The aim is to develop a simple and computationally cheap halogen chemistry scheme which is able to reproduce most of the effects of halogen species on climate-relevant species, i.e. ozone, methane via oxidizing capacity, and then explore how halogen chemistry can affect the sensitivity of ozone (and the related oxidizing capacity) to its precursors, a potentially important considerations in chemistry-climate interactions. The challenge is to find the right compromise between the level of realism and details (e.g. species and processes) acceptable for the chemistry-climate applications, and the computing efficiency for the long chemistry-climate simulations. Therefore, we only represent the reactions and species which are dominant for the impact of halogens on global ozone destruction. The impacts of halogen chemistry on key short-lived climate forcers (such as ozone and methane) are still totally ignored in the multi-model chemistry-climate simulations. It is probably time for these models to include some of these effects.

The objective of the paper is to explain the methodology adopted, the potential and limitations of such a simplified model. It is all new and valuable information for chemistry-climate modellers. As such, the paper is well suited for GMD.

Among the list of manuscripts published by GMD are:

- geoscientific model descriptions, from statistical models to box models to GCMs;
- development and technical papers, describing developments such as new parameterizations or technical aspects of running models such as the reproducibility of results;

In addition, we show how iodine, bromine and chlorine chemistry can, beyond their already demonstrated impact on tropospheric ozone, also affect the sensitivity of ozone to its precursors, a potentially important considerations in chemistry-climate interactions. These are novel results.

We recognize that the language was sometimes vague and some formulations were clumsy. We sincerely apologize and thank the reviewer for his/her recommendation. We did a thorough work on the manuscript in order to clarify the intent and better organize it to make

the flow more logical. Below we give more details on how we responded more specifically to these general and further specific comments.

Specific major comments are detailed below:

 L35 - VSLS are not the only source of reactive halogens, this is a mistake. Halogens are also released from sea salt aerosols and iodine is released via ozone deposition to the ocean surface. These should be discussed and included in the modeling. These are mentioned later (around L135), but this must be clarified here. Polar specific halogen activation mechanisms (snow, blowing snow sea salt aerosols, etc) may not need to be included here for the first global modeling study with LMDZ-INCA, but should at least be mentioned.

We agree, it has been added to the introduction. Note that iodine release from the ocean is considered via prescribed fields.

L101 - The heterogeneous chemistry on sea salt and sulfate aerosols needs to be more clearly detailed.

The list of reactions has been added to make the paper more descriptive.

 The basic aerosol chemistry within LMDZ prior to adding this must be described in some detail in order to know if what has been added is reasonable. The aerosol chemistry in the two established global models (CAM-Chem and GEOS-Chem) are significantly different. The method implemented here follows the CAM-Chem methodology.  However, this implementation seems to only include recycling on sea salt and sulfate aerosols, rather than actual degassing of HCl and HBr triggered by aerosol acidification (or another proxy for acidity, for example sea salt aerosol age). This approach for reacative halogen cycling is already used in CAM-Chem, but it is not correct compared to what is known to happen in the atmosphere. This is particularly important for chlorine release from sea salt aerosols globally. One early example of modeling HCl emissions from sea salt aerosols is:

Erickson III et al., JGR, 1999:
https://agupubs.onlinelibrary.wiley.com/doi/abs/10.1029/98JD01384

Other processes that liberate photochemically active halogens from sea salt aerosols are also known, see for example the early review of Finlayson-Pitts and Hemminger, J. Phys. Chem. A, 2000:

https://doi.org/10.1021/jp002968n

For this first implementation of halogens chemistry in INCA, we did some trade-offs and incorporated a simplified representation of heterogeneous chemistry on sea-salts and sulfates which are prescribed. We added the chemical scheme to the main manuscript to help the reader to clearly understand what is done and clarified the text. We also clarified in part 2.1.1. that the version of INCA used here does not compute aerosols interactively, they are specified which limits the level of sophistication of the scheme.

L105: I do not understand the discussion of ice chemistry here, is this ice/mixed phase clouds, snow, or sea ice. What does it mean to ensure bromine recycling in the MBL?

We apologize for the lack of clarity here. We mean reaction on ice crystals in clouds. We modified the sentence here and remove the confusing sentence about ensuring recycling in the MBL.

L133: How is HCl emitted from sea salt?

We use the geographical distribution of sea-salt and impose the total HCl emitted to match the one reported by Hossaini et al. 2016.

L181 - Section 3.1.1 - This section is poorly written and I cannot tell what is being compared with what. There is not enough information or numbers in this section for this to be considered a chlorine model evaluation. The information in this section should be provided in figures, such as a plot of total Cly in LMDZ-INCA compared with CAM-Chem or GEOS-Chem for the same years/conditions as a function of latitude and altitude.

We rewrote this section to clarify what is compared with what. However, the chlorine chemistry is not central as we first aim to represent correctly the ozone loss, which mainly driven iodine and bromine chemistry. We have thus modified the flow of the paper to make this simplified evaluation of halogens concentrations more logical in the context of the paper and its aim.

L201 - Section 3.2.1 - Same comments as for Section 3.1.1. (above).

In addition, the discussion of wanting to capture the impact on ozone rather than the actual chemistry is a puzzling and problematic argument. The focus on ground level ozone, while also discussing the entire tropospheric chemistry cycle for ozone at the same time is also a problem.

We now better explain the strategy in the introduction: what is the most important according to previous studies to represent the effect of halogenated chemistry on global tropospheric ozone and thus better hierarchize the evaluation on elements which are of importance for the ozone budget.

L218 - Section 3.1.3 - Same comments as for Section 3.1.1 (above). I don't understand why there is a separate iodine flux section, while the bromine and chlorine fluxes are not discussed. I believe this flux section is actually a chemical budget, but the writing is unclear.

We now try to better guide the reader.

L247 - Industrial emissions of chlorine are known to be important for ground level reactive chlorine cycling (see Thornton et al., 2010, https://www.nature.com/articles/nature08905). Why are they left out here?

We chose to follow the recommendations of Wang et al. (2019) and not to include anthropogenic emissions in our reference simulation since the total industrial emissions constitute only a very small fraction of the total HCl emitted on a global scale (6700 Gg Cl.yr-1 < 100000 Gg Cl.an-1).

L252 - I don't understand how the measurements and model are compared in Table S9. Where are the measurement sites, what model grid cells are used? What does it mean that the measurements are a 24 hour average while the model uses an average over months?

Measurements correspond to average of several 24hours means measured over 1 or 2 months. This is clarified in the legend of the Table (now S3). These results have been aggregated by Wang et al. 2019, we use them in a similar way.

L255 - Table 2 - I understand that the model values are diurnal averages, but more is needed on the measurements (timing of measurement). Are you comparing a monthly average from the observation in some cases with a diurnal average from the model? The wording is unclear and it is possible that the model and observations cannot be compared in this way.

Not applicable, the Table has been removed.

L269 - Section 4 - This section shows why the discussion earlier regarding the main goal of the study is to capture ozone chemistry near the surface is not accurate. In this section the whole troposphere is discussed (however this is not well defined).

We clarified the aim which is not the surface but the troposphere.

L289 - The discussion of the factors that result in different tropospheric ozone burdens from different NMVOC emissions needs to be expanded and discussed in more detail. Is the difference due to NMVOC emissions even larger than due to inclusion of halogen chemistry? If yes, then why were the same NMVOC emissions not used for these runs?

This discussion has been removed as it didn't bring valuable information here.

L371 - Section 5 - I do not see any new conclusions compared to past work. This should be improved. The main problems are that (1) the modeling is not advanced compared to past work and (2) there are no new conclusions. Therefore, this paper does not fit the aims of GMD at present. However, maybe the GMD editorial team can agree to accept new descriptions of models without these two criteria being met. In general, it is very good to see more models including halogen chemistry. Therefore, it would be a dissapointment if this review process ended with the result that there is no halogen chemistry available in LMDZ-INCA in the future.

We did not state clearly that the aim was not to improve the pre-existing halogen chemistry schemes or carry out a process study about halogen chemistry. The aim is a simplified and cheap scheme which is able to reproduce most of the effects of halogen species on climate-relevant species, namely ozone, for chemistry-climate applications. This paper effectively does not bring new conclusions regarding detailed halogen processes as the aim is to show that the simple INCA scheme is able to "mimic" the few other complex models regarding the impact of halogens on global ozone. Considering how complex is the implementation of halogen chemistry, we consider it as an important milestone for the model. The results about the impact of halogen chemistry on the sensitivity of ozone to its anthropogenic precursors are also new and relevant for chemistry-climate interactions. It shows that a simplified model can also be valuable depending on the application. We understand from the GMD

 that such a paper fits within their scope. All this information will certainly encourage other global chemistry-transport and chemistry-climate models to pursue such implementation and explore other chemistry-climate interactions involving halogens. If only halogen schemes that are more advanced (and so more costly computationally) than the few existing ones can be published now, we do not know how the long-standing issue about not accounting the impacts of halogen chemistry on key short-lived climate forcers (such as ozone and methane) in all the multi-model climate studies will be resolved.

Comments regarding citations, co-authorship, and code availability:

- There is a persistent issue of self citations within the paper, while ignoring the wider literature. Examples are on lines 33 and 45, only including self citations for the authors' own groups is not appropriate for these sentences.

We now mention the 3 models having incorporated halogen chemistry and cite more studies done by these models in the introduction.

- For co-authorship - the description of the author contributions is interesting, but may not rise to the level of the GMD co-authorship guidelines (https://www.geoscientific-model-development.net/policies/obligations_for_authors.html, point 9). Specifically, serving as a scientific advisor for the manuscript may not fit within this description from GMD.

The description was clumsy, contributions have been clarified and fulfil the GMD standards.

The code availability on google drive is not standard and this code should be archived on GitHub and Zenodo as noted by the GMD executive editor.

A doi to distribute the code and data will be created if the paper is accepted for publication

Minor comments:

L19 - halogens should not have an s here

done

L40-41 - halogens both change the partitioning of NOx and also react with NO2 to form species that remove NOx from the system via deposition or other loss processes. While this is said in the next paragraph, this statement is not quite complete/right here.

"and also react with $NO_2$ to form species that remove NOx from the system via deposition or other loss processes" has been added

L184 - halogens' should be halogen

done

---

## Referee Report (RR1)

**Review of : Sensitivity of tropospheric ozone to halogen chemistry in the chemistry-climate model LMDZ-INCA vNMHC by Caram et al., 2022.**

This a resubmission of the original manuscript that was assessed by two previous independent referees. There was a general agreement between referees that the heterogeneous component of the model is lacking some key reactions, which are important for recycling and release/re-cycling of halogen species back into the gas-phase of the troposphere allowing photo-chemical destruction of e.g. Ozone ($O_3$) and methane ($CH_4$). Thus the accurate simulation of the mixing ratios of Cl, Br and I radicals is a prerequisite towards capturing the effects well with respect to climate impacts. Both of the previous referees suggested a major revision in order to achieve the quality necessary for publication in GMD. Looking at the modifications made to the manuscript I am not sure that issues have been fully addressed from the previous versions. There is the danger of formulating a scheme which doesn't simulate enough halogen radicals which, when applied in e.g. a chemical ensemble to predict $CH_4$ lifetimes, would increase uncertainty in the resulting projections due to the lack of key processes. The authors even admit that the simulated Chlorine component is not optimal, although both Bromine and Iodine have higher chemical destruction fluxes against the main tropospheric reactants therefore the under-estimate in total oxidative capacity is likely low. The evaluation of the scheme against observations is very weak and, this being a GMD development paper, needs addressing before publication to meet the requirements of the journal.

Major comments:

(i) Climate and oxidative capacity is moderated by photolysing light, which itself is attenuated by scattering processes involving clouds and aerosols. A large fraction of $CH_4$ is oxidised in the tropical region, which exhibits lower $O_3$ mixing ratios in the lower atmosphere than the more polluted Northern mid-latitudes meaning that the destruction via halogen radicals becomes most important. Sulphate production in clouds is enhanced by the oxidation in droplets via HOCl(aq) and HOBr(aq), but this process is missing from the update whereas there is a direct link to process important for determining future RF (with the S component in more remote regions being biogenic in origin via DMS oxidation). With low $H_2O_2$ in more remote locations, the halogen-sulphate production route could determine AOD over the oceans. Can the authors comment on why this omission was made considering the proposed application of LMDZ-INCA towards chemistry-climate studies.

(ii) The main acid catalysed routes are missing from the aqueous phase chemical scheme applied in the model, as acknowledged by the authors, but should be accounted for in some way. If the authors do not wish to implement a complex heterogeneous scheme they should apply a first-order generation rate to account for release of precursors e.g. -> $Cl_2$(g) in $s^{-1}$ (based on available cloud and/or aerosol SAD as a meteorological parameter for parametrizing the extent of activation). This could also be done for BrCl(g) release. This could then act as a guide (and reference) for a scheme applicable in other chemistry-climate models rather than a more simplistic version of a CTM scheme. The influence of applying such an assumption should be

shown by comparing two simulations with respect to the tropospheric Cl burden, with and without such reactions active.

(iii) Section 3 : Both the total burdens in Sherwen et al. (2015) and Wang et al. (2021) should be put into the text and tables to show the underestimation in tropospheric Cl burden due to the omission of the heterogeneous chemistry discussed above. This avoids undermining the main conclusion in Wang et al., (2021) in that a complex description of heterogeneous chemistry is necessary for an accurate tropospheric Cl burden. A much more important inclusion would be values from a new run using the assumed release rates of halogen precursors.

(iv) Figure 2 shows the latitudinal and zonal mean impact on tropospheric $O_3$. There is no proof this moves the model towards a better description of the distribution of $O_3$ in the global troposphere. There are $O_3$ profile measurements readily available e.g. at the south pole which could be used to validate the substantial percentage reductions in $O_3$ burden for these latitudes. Currently the reader has no idea as to whether the inclusion of halogen chemistry is needed in this version of LMDZ (which would increase resources needed in any long-term run) or what the potential biases with and without are meaning confidence in the model capturing the correct global $O_3$ distribution is low. I find this not acceptable for a GMD paper, which doesn't need new scientific findings but rather some evaluation of the update to show the model performs well.

Minor comments

(i) When multiple references are used they should always be given in a chronological order throughout the text e.g. ln 28.

(ii) Most of the reaction rates originate from the standard recommendations apart from some exceptions. For instance, Cl + $C_2H_4$ has a IUPAC recommendation but a single study value (which may not be the optimal choice) is applied. Why not use a full set of rate data from the recommendations? The IUPAC reaction for Cl + $C_2H_4$ is defined as a third-body reaction, whereas the rate applied here is second-order only. Although the impact of global modelling of $O_3$ will be small, a correct description of rate data should be applied.

(iii) There are no references for the assumed uptake parameters given in Table 5, even though many of these parameters have been measured and are available in the recommendations. This introduces a discontinuity in the basis upon which the gas-phase and aerosol-phase chemistry is sourced. The uptake values for various species currently applied are identical for both sea-salt and sulphate aerosol which doesn't seem to be correct. Sea-salt is typically 'wet' considering the Relative Humidity over oceans, with IUPAC providing e.g. an uptake value of 0.6 for HOBr higher than that used here, whereas for 60%wt $H_2SO_4$ it reduces to 0.06. Similar differences can be found for other halogen species. This will alter the global chemical budget terms provided.

---

## Referee Report (RR2)

Review of Caram et al., 2022

Caram et al report results from the LMDZ-INCA model after adding tropospheric reactive halogen chemistry. The chemical mechanism mainly includes gas-phase chemistry, and thus lacks some heterogeneous chemistry that has been demonstrated to be important for reactive halogen production and loss. Nonetheless, this represents an important step forward for a global chemistry-climate model. Tropospheric reactive halogens have been shown to be an important lever for controlling the oxidation capacity of the atmosphere, so chemistry models should include this chemistry. The chemistry is complex and has large impacts on ozone, making it a somewhat scary endeavor. Thus, I commend the authors for taking this step forward in their model.

They mainly compare their model with other models that have a more complex representation of tropospheric reactive halogen chemistry, CAM-Chem and GEOS-Chem. Despite lacking important heterogeneous chemistry, their model agrees reasonably well with these other models, with some discrepancies that they attempt to explain. Results comparing preindustrial and industrial simulations and the impact of reactive halogen chemistry during these two time periods also is similar to these other models.

My main suggestions are to clarify the chemistry that is included in their model. Specific examples are listed below:

- In Figure 1, does "chlorine contribution to bromine recycling" indicate the three reactions above this statement, or other reactions? If the latter, which reactions are these? Are they in Table 5?
- Also in Figure 1, "heterogeneous reactions of bromine on sea salt and sulfate aerosols" is stated, but it would be helpful to refer to specific reactions in the tables. I don't see these reactions in Table 5.
- Table 5 lists the reaction probability for heterogeneous reactions. The reaction probability formulation is typically used for first order reactions, but some reactions listed here are second order, so I'm confused. More detail is needed on how the reaction probability is used for these second-order reactions.
- Sea salt chloride is emitted at HCl. Why is sea salt bromide not emitted as HBr? Sea-salt bromide is the largest primary source of bromine to the troposphere.
- Figure 5 says that reactive bromine is produced from N2O5, but I don't see this reaction in Table 5.
- It is mentioned in the text that N2O5 does not produce ClNO2, but this is inconsistent with Table 5.

---

## Referee Report (RR3)

**Review of the manuscript "Sensitivity of tropospheric ozone to halogen chemistry in the chemistry-climate model LMDZ-INCA vNMHC" Caram et al., GMDD, 2023.**

The paper describes the implementation of a halogen chemistry scheme into the LMDZ-INCA model, which is based on previous developments of short-lived chlorine, bromine and iodine chemistry in CAM-Chem and GEOS-Chem. This is of major importance, as tropospheric halogen chemistry has been shown to influence the ozone budget as well as the methane burden, and many works have highlighted the importance of including this chemistry in future chemistry-climate projections. Even though the halogen chemical scheme LMDZ-INCA does not contain the full scheme presented in other models, the authors have selected the main reaction and processes as a compromise between detailed chemistry and computational costs during this initial implementation, which can be updated in future steps as the model is used for climate studies. Despite the paper focuses on the technical implementation, they provide a very interesting evaluation of how sensitive is the global tropospheric ozone budget to perturbations in the emissions of air-pollutants, comparing results for different configurations including and neglecting the contribution of chlorine, bromine and iodine; as well as a comparison of results for pre-industrial and present-day scenarios.

The paper is well organized and provides details on the new processes that are currently implemented in LMDZ-INCA, although there are several technical details and/or inconsistencies that are not clear on some of the tables, figures and descriptions presented in Sections 2 and 3. Thus, and even though I do not find any relevant scientific issues that must be solved before publication, given the large number of technical details that must be corrected and/or clarified in the manuscript, I would like to see them implemented in a revised version of the manuscript before publication. I hope the detailed list provided below helps the authors to finalize the description of the model.

**Main Comments:**

P4, L122: Following other reviewer's request during previous revisions, the authors explicitly state that *"Recycling through heterogeneous chemistry on sea salt and sulfate aerosols is included while the heterogeneous bromine reactions in water droplets and on ice crystals in clouds are not"*. Even though this is fine for the initial and simplified halogen implementation in LMDZ-INCA, something should be mentioned (either here or below in the text where

sources are described) that the model does not include any type of inorganic bromine source from sea-salt aerosols, while in CAM-Chem and GEOS-Chem this is the dominant bromine source to the troposphere. I also suggest this issue to be mentioned in the discussion and/or conclusions when the range of impacts of halogens on tropospheric compositions is quantified. In relation to this, P15, L321-323: It is expected that tropospheric bromine burden in LMDZ-INCA is in the low range of other models and observations due to the missing source of SSA-recycling. Later on L330, you indicate that *"Our choice to include, in this first step, reactions on sea salt and sulfate aerosols only is …"* but the bromine heterogeneous reactions you mention here do not represent a net bromine source from SSA. Please make this clarification explicit to avoid conclusion.

P10, L184-187: The authors explicitly state that: *"The inorganic iodine distribution of emissions (HOI and I2) are prescribed from Prados-Roman et al., (2015a) …"*. Based on this, I understand that an emissions file is read off-line from the model instead of being computed on-line as in CAM-Chem. If this is the case, this should be explicitly indicated. In addition, some discussion about how these "prescribed" emissions are considered for pre-industrial conditions should be provided, because for example, changes of surface ozone between pre-industrial and present-day in CAM-Chem and LMDZ-INCA differ. How these changes translate into changes in iodine emissions in LMDZ-INCA? In line L198 you state that emissions are 40% lower. Is this based on INCA O3 changes or taken from Prados-Roman et al.?

**Minor Comments:**

P2, L46: In addition to Iglesias-Suarez et al., 2020, another study from the group focused on halogen influence on ozone budget and the oxidative capacity of the atmosphere is Badia et al, 2021. This paper should be cited when the budget results presented in Table 7 are discussed.

P2, L58-74: When the discussion about the compromise between detailed atmospheric chemistry mechanisms vs. computational costs is presented, the authors could refer to Fernandez et al., 2020, where Surrogate, Explicit and Full halogen schemes for bromine are compared.

P3. L105: Are biogenic emissions for pre-industrial times identical to present? Please clarify.

P3, L110: Authors should add a reference for HTAP.

P9, L154: The resolution is longitude x latitude, or vice versa? Also, 39 levels up to which height?

P9, L157: All simulations used the same meteorology from ECMWF, including the pre-industrial ones? This is common procedure in many other modeling works, please make sure to make it explicit.

P9, L161: Could you provide at least in the text the CH4, NOx and NMVOCs emission values for present-day and pre-industrial conditions? Absolute values are necessary to compare with other models and to estimate how much represents -20% change.

P9, L171: I can imagine that CH2BrCl, CHBr2Cl and CHBrCl2 were not included due to the compromise between computational cost and number of species/reactions. But as those species are included in the original Ordoñez et al., 2012 inventory, at least something should be said about them.

P11, L221-223: I would expect that the ozone percentage changes in the SH are larger because, at least partially, the absolute abundance are smaller.

P12, L248: I suggest pointing at the individual reactions numbers in each table.

P12, L255: Please indicate how the ozone lifetime is computed (e.g., burden / losses?).

P13, L263: Here you mention (but do not define) X*, which is not used until L360, where it is only defined for Chlorine. Please clarify its definition and usage.

P13, L272: Why do you exclude IBr, ICl and aerosols for the burden computation?

P14, L298-299: The implementation of HOI and IONO2 recycling on ice-clrystals in Saiz-Lopez et al. 2015 allowed to reproduce IO upper tropospheric abundance by "increasing" Iy lifetime against washout. I would expect including those reactions here will even increase the LMDZ-INCA IO overestimation. Can you comment on this?

P15, L321: There is a misunderstanding of the captioned values: Both (2.0 +/- 1.5 pptv) and 3.2 +/- 1.6 pptv) BrO values are observations at different heights performed by Dorf et al. (2008) in the tropical tropopause, which Fernandez et al. (2014) indicates are in agreement

with CAM-Chem modeling results. But those are observations, not modeling results. Please correct.

P19, L416: The emissions of photo-oxidant precursors has been perturbed by 20% compared to present-day values, isn't it? Please make it clear. Why did you chose to reduce instead of increase by 20% the air-pollutants emissions?

**Figures and Tables**

Table 1: I found a few inconsistencies between the table and the text, as well as to how LMDZ-INCA has mapped the halogen chemistry scheme from CAM-Chem. In particular:

T1a: Methyl Bromide (CH3Br) emissions are not considered in CAM-Chem, this model uses a LBC approach as for Long-lived halogens. Also, I'm surprised that Bromoform (CHBr3) is omitted in the table. Is there a typo or an omission here?

T1b: HCl in LMDZ-INCA is emitted both from the surface as well as from sea-salt, isn't it? Then both circle and square symbols should be included.

T1c: I'm surprised that dry-deposition is considered for VSL compounds like CH2IBr and CH2ICl. Why is that and following which model? Why this is not considered for other VSL compounds like CH2Br2. Is this a typo?

Table 2: The title of the table contains too much information that would much better fit as a table footnote. In particular, reference to J29, J30, and quantum-yields 2 and 30 are mentioned, but no number assignation for any of the photolysis reactions is given. Please make it consistent. Also, I noticed there is no photolysis of HCl and HBr, while this process is considered for HI. Could this be affecting the partitioning of chlorine species where, for example, HCl is the most abundant of all chlorine reservoirs?

Table 3: Please move the detailed information from the table title to an appropriate and well referenced footnote (e.j., define A0 and Ea). Replace reation by reaction.

Table 4: Please update table title and footnotes as for previous tables.

Table 5: Please define I2Ox. Also note that footnote c is applied only to the first reaction on the table, although I believe it corresponds to all reactions with 2 reactants.

Table 6: Please move details from the title to footnotes as for previous tables. Most importantly, note that emission values are provided in Gg yr-1, and not Tg an.-1. In particular, the HOI-I2 emissions published in Prados-Roman et al. (2015) is 1.9 Tg I yr-1, where in your table it is aprox. 2.5 Tg yr-1. How can you explained those differences if emissions, as stated above, are prescribed to Prados-Roman et al.? Have the original Prados-Roman emissions been scaled to surface O3 deposition in LMDZ-INCA? Finally, bromine emission from sea-salt increases from PI to PD, while for the case of HCl it remains constant. Why is that? Is this due because acidification is not considered?

Table 7: Based on the numbers compiled in the table, the halogen impact over ozone seems to be larger than for other models (e.g., Badia et al., 2021). Indeed, I believe the ozone burden for MLDZ-INCA are lower than for CAM-Chem and GEOS-Chem. I'm more familiar with positive STE values, why do you present them as negative?. Anyhow, your results show that in absolute terms STE is smaller when halogens are considered … Which makes me ask: Does LMDZ-INCA include a prescribed representation of the stratosphere? Or is the stratosphere chemically-coupled to the troposphere? P12, L239-240: I would expect that the different coupling to the stratosphere between models also affects results here.

Table 8: Given that the implementation of halogen chemistry is based on CAM-Chem, I recommend to include methane lifetimes values from Li et al., (2022) in the table.

Figure 4: The caption indicates that numbers in boxes represent the mass balance of the species, but does not explain what the numbers above each colored arrow are.

Figure 5. *"Percentages in brackets represent the branching ratios"*. I do not understand what the values in brackets are, not how they are computed.

Figure 7: I like a lot the schematic representation of how halogens modify the ozone budget for pre-industrial and present day conditions. I suggest including in parenthesis the percentage change of each of the reported burden changes printed on the side of each arrow.

Supplement Header: Affiliations are missing. They should be made consistent with the main text.

Figure S2: Please indicate a reference for the HCl observations, and if possible, a range of values. Do not use capital letters for "Mixing Ratio" and "pptv" in the title.

Table S2: Reference for VSL chlorocabon emissions in Table S2 are incorrect, at least for CAM-Chem. None of the papers cited in footnote considered anthropogenic VSL chlorocarbons. Note that the Ordoñez inventory only consider natural oceanic emissions.

**Language editing comments and Typos:**

P1, L14: I suggest replace with "… and evaluate halogen effects on the tropospheric ozone burden".

P1, L24: I was surprised the first sentence of the introduction started with "If …". I suggest rephrasing unless strictly necessary.

P1, L34: "oxide radicals"

P3, L83: "… and their effect on ozone by using …"

P4, L122: Replace "modellers" by "other models".

P9, L195-196: Revise grammar when defining CH3Br LBCs for pre-industrial periods.

P10, L207: "simulating the climatology of halogen species".

P11, L213: "which include Cl, Br and I"

P11, L236: Ox = Odd Oxygen.

P14, L293-295. Please revise the sentence grammar and if there is something missing.

P19, L421: "the chemical lifetime of CH4 reflects the lifetime of CH4 to removal by OH". Please revise.

P20, L452: "CO and NMVOC"

Fig. S1: missing →mixing

**References**

Many references and citations include a,b letters that are not correct (e.g., Fernandez et al., 2014b; Ordoñez et al., 2012a,b). See for example (P3,L13). This should be corrected all over the manuscript.

P2, L44; P2, L55; P3, L113 and elsewhere: Note that sometimes citations in the text are in chronological order, and some others in alphabetical order. Please follow the journal guidelines.

P9, L181: Check parenthesis in citation to Schmidt et al., 2016.

P22, L531: Fernandez et al., 2014b is incorrect, it corresponds to Saiz-Lopez et al., 2014.

P24, L611: Ordoñez et al., 2012 is duplicated.

Badia, A. et al. The Role of Natural Halogens in Global Tropospheric Ozone Chemistry and Budget Under Different 21st Century Climate Scenarios. J. Geophys. Res. Atmos. 126, e2021JD034859 (2021).

Dorf, M., et. Al,: Bromine in the tropical troposphere and stratosphere as derived from balloon-borne BrO observations, Atmos. Chem. Phys., 8, 7265–7271, doi:10.5194/acp-8-7265-2008, 2008.

Fernandez, R. et al., Bromine partitioning in the tropical tropopause layer: implications for stratospheric injection. Atmos. Chem. Phys. 14, 13391–13410 (2014).

Fernandez, R. et al. Intercomparison Between Surrogate, Explicit, and Full Treatments of VSL Bromine Chemistry Within the CAM-Chem Chemistry-Climate Model. Geophys. Res. Lett. 48, e2020GL091125 (2021).

Iglesias-Suarez, F. et al. Natural halogens buffer tropospheric ozone in a changing climate. Nat. Clim. Chang. 10, (2020).

Li, Q. et al. Reactive halogens increase the global methane lifetime and radiative forcing in the 21st century. Nat. Commun. 13, 2768 (2022).

Ordóñez, C. et al. Bromine and iodine chemistry in a global chemistry-climate model: description and evaluation of very short-lived oceanic sources. Atmos. Chem. Phys. 12, 1423–1447 (2012).

Prados-Roman, C. et al. A negative feedback between anthropogenic ozone pollution and enhanced ocean emissions of iodine. Atmos. Chem. Phys. 15, (2015).

Saiz-Lopez, A. et al. Iodine chemistry in the troposphere and its effect on ozone. Atmos. Chem. Phys. 14, 13119–13143 (2014).

Saiz-Lopez, A. et al. Injection of iodine to the stratosphere. Geophys. Res. Lett. 42, (2015).

---

## Author Response (AR2)

The new version of your manuscript is much improved; however the second round of review raised some issues that should be addressed before publication. Please consider the suggestions of reviewer #1, which are worth being tested and could lead to an improvement in your halogen extension. Also, since you stated that improving ozone is the main objective, more effort should be put into the validation part its evaluation.

Dear editor,

The main reproach of the reviewer claiming for major revision is the lack of consideration of a major source of chlorine in the model and the need of a representation, even if coarse, of this source. Actually, this is a misunderstanding since this source is represented even if its dependency to pH is not accounted for. The justification of this choice was detailed in the answer to the first set of reviews and clarified in the text at this time.

The second point, that you underline, is the lack of comparison between model outputs and vertical sonde measurements for ozone concentrations. Here we attempt to include the representation of a complex chemistry, for which some chemical processes, sources and sinks are still under debate in the literature to see how it affects the sensitivity of ozone. The aim of a global chemistry transport model can be to reproduce concentrations but above all their aim is to reproduce the sensitivity of species of interest for air pollution or climate issues to change, for example, in anthropogenic emissions (or climate). A model can have a systematic bias in representing some concentrations but can be fitted for purpose, for example can allow to assess change in composition due to human activity as discussed in the Box 6.1 in the chapter 6 of the WG1 IPCC report (2021). Comparisons between models and observations are necessary and can inform on missing processes, but not everything can be assessed through model-observation comparisons and in particular here by doing comparisons between sonde measurements at specific locations and global model outputs due both to representativity difficulties and systematic biases already identified to represent ozone in extratropical troposphere in the southern hemisphere. A systematic benchmarking of this new versions of the model is beyond the scope of this paper but is systematically realized when such a model takes part to international intermodel exercice.

**Review #1**
This a resubmission of the original manuscript that was assessed by two previous independent referees. There was a general agreement between referees that the heterogeneous component of the model is lacking some key reactions, which are important for recycling and release/re-cycling of halogen species back into the gas-phase of the troposphere allowing photo-chemical destruction of e.g. Ozone ($O_3$) and methane ($CH_4$). Thus the accurate simulation of the mixing ratios of Cl, Br and I radicals is a prerequisite towards capturing the effects well with respect to climate impacts. Both of the previous referees suggested a major revision in order to achieve the quality necessary for publication in GMD. Looking at the modifications made to the manuscript I am not sure that issues have been fully addressed from the previous versions. There is the danger of formulating a scheme which doesn't simulate enough

halogen radicals which, when applied in e.g. a chemical ensemble to predict CH4 lifetimes, would increase uncertainty in the resulting projections due to the lack of key processes. The authors even admit that the simulated Chlorine component is not optimal, although both Bromine and Iodine have higher chemical destruction fluxes against the main tropospheric reactants therefore the under-estimate in total oxidative capacity is likely low. The evaluation of the scheme against observations is very weak and, this being a GMD development paper, needs addressing before publication to meet the requirements of the journal.

We thank the reviewer for his/her review, please find hereafter our responses. Note that if some modifications asked by previous reviews were not done this is because there was a misunderstanding about what is or not considered in the model and the limitations of the scheme due to these hypotheses (compromise between complexity and CPU+skill of the model). Explanations were given in the reply to the first set of reviews and clarification had been done in the text to address them.
Major comments:

(i) Climate and oxidative capacity is moderated by photolysing light, which itself is attenuated by scattering processes involving clouds and aerosols. A large fraction of CH4 is oxidised in the tropical region, which exhibits lower O3 mixing ratios in the lower atmosphere than the more polluted Northern mid-latitudes meaning that the destruction via halogen radicals becomes most important. Sulphate production in clouds is enhanced by the oxidation in droplets via HOCl(aq) and HOBr(aq), but this process is missing from the update whereas there is a direct link to process important for determining future RF (with the S component in more remote regions being biogenic in origin via DMS oxidation). With low H2O2 in more remote locations, the halogen-sulphate production route could determine AOD over the oceans. Can the authors comment on why this omission was made considering the proposed application of LMDZ-INCA towards chemistry-climate studies.

The aim of this version of the model is to simulate the gaseous chemistry and in particular the oxidizing capacity. Aerosols and in particular the effect of halogenated chemistry on AOD through in-cloud chemistry is beyond the scope of the paper. even if, as underlined by the reviewer, this effect is relevant to determine the total effect of halogenated chemistry on climate forces (and finally on climate). Here, we do not use such a model to compute future RF but Earth System Model. Not all models can include every single process, it depends on the application of the model. As explained in the introduction of the paper, for the moment, none of the models used to assess present and future ozone effect on climate in CMIP includes halogenated chemistry, and very few chemistry-transport models include it to quantify the role of this chemistry. This work is a first important step before adding more complexity.

(ii) The main acid catalysed routes are missing from the aqueous phase chemical scheme applied in the model, as acknowledged by the authors, but should be accounted for in some way. If the authors do not wish to implement a complex heterogeneous scheme they should apply a first-order generation rate to account for release of

precursors e.g. -> Cl2 (g) in s -1 (based on available cloud and/or aerosol SAD as a meteorological parameter for parametrizing the extent of activation). This could also be done for BrCl(g) release. This could then act as a guide (and reference) for a scheme applicable in other chemistry-climate models rather than a more simplistic version of a CTM scheme. The influence of applying such an assumption should be shown by comparing two simulations with respect to the tropospheric Cl burden, with and without such reactions active.

The reviewer is right: the acid catalysed route is a major source of reactive chlorine that can not be neglected and has to be considered even if in a very simple way. Actually, we do not neglect this source, it is represented even if it is done in a coarse manner, which is not dependent on the pH. We chose to apply the same methodology as in Hossaini et al . 2016.

(iii) Section 3 : Both the total burdens in Sherwen et al. (2015) and Wang et al. (2021) should be put into the text and tables to show the underestimation in tropospheric Cl burden due to the omission of the heterogeneous chemistry discussed above. This avoids undermining the main conclusion in Wang et al., (2021) in that a complex description of heterogeneous chemistry is necessary for an accurate tropospheric Cl burden. A much more important inclusion would be values from a new run using the assumed release rates of halogen precursors.

The total tropospheric burden is 290Gg of HCl versus 306 Gg in Wang et al. 2021. As far as we can see, the burden found by Sherwen et al. 2015 is not given in their paper. The difference is not due to omission of source but to the fact that we use HCl emission from Hossaini et al. 2015 which is a little lower than the source considered in Wang et al. 2021. This difference is about 5% which is low considering the uncertainty in chlorine budgets.

 (iv) Figure 2 shows the latitudinal and zonal mean impact on tropospheric O3 . There is no proof this moves the model towards a better description of the distribution of O3 in the global troposphere. There are O3 profile measurements readily available e.g. at the south pole which could be used to validate the substantial percentage reductions in O3 burden for these latitudes. Currently the reader has no idea as to whether the inclusion of halogen chemistry is needed in this version of LMDZ (which would increase resources needed in any long-term run) or what the potential biases with and without are meaning confidence in the model capturing the correct global O3 distribution is low. I find this not acceptable for a GMD paper, which doesn't need new scientific findings but rather some evaluation of the update to show the model performs well.

The aim of this work is to better represent the chemistry of the ozone system and its sensitivity to precursors. Chemists know that the halogen chemistry occurs and has a significant impact and is thus relevant to implement in global models. However, global models already underestimate systematically the ozone concentrations in the southern lower troposphere (see Griffiths et al. Figure 3, right panel comparing ozone data and multimodal results in the South Pole and showing a systematic underestimate of about

27%) which implies that the inclusion of halogenated chemistry does not improve the model results. It does not mean that halogenated chemistry does not take place or leads to a worse representation of the chemistry but that neglecting it compensates partly for a systematic problem in global models. The uncertainty in representing ozone by global models in this region exceeds 30% (see Griffith et al. Figure 2 top panel on the right). The aim of a global chemistry transport model can be to reproduce concentrations but, above all, their aim is to reproduce the sensitivity of species of interest for air pollution or climate issues to change, for example, in anthropogenic emissions (or climate). A model can have a systematic bias in representing some concentrations but can be "fitted for purpose", for example can allow to assess change in composition due to human activity for example as discussed in the Box 6.1 in the chapter 6 of the WG1 IPCC report (2021). A systematic benchmarking of this new versions of the model is beyond the scope of this paper but is systematically realized when such a model takes part to international intermodel exercice.

Minor comments (i) When multiple references are used they should always be given in a chronological order throughout the text e.g. ln 28.

The references are given by alphabetical order and then by chronological order for multiple papers with the same first author. The compliance with GMD citations rules will be ensured.

 (ii) Most of the reaction rates originate from the standard recommendations apart from some exceptions. For instance, Cl + C2H4 has a IUPAC recommendation but a single study value (which may not be the optimal choice) is applied. Why not use a full set of rate data from the recommendations? The IUPAC reaction for Cl + C2H4 is defined as a third-body reaction, whereas the rate applied here is second-order only. Although the impact of global modelling of O3 will be small, a correct description of rate data should be applied.

We thank the reviewer for this notification. The value will be revised in the next version. As mentioned by the reviewer this reaction is not significant and, actually, is not considered in the other models accounting for the chlorine chemistry.

(iii) There are no references for the assumed uptake parameters given in Table 5, even though many of these parameters have been measured and are available in the recommendations. This introduces a discontinuity in the basis upon which the gas-phase and aerosol-phase chemistry is sourced. The uptake values for various species currently applied are identical for both sea-salt and sulfate aerosol which doesn't seem to be correct. Sea-salt is typically 'wet' considering the Relative Humidity over oceans, with IUPAC providing e.g. an uptake value of 0.6 for HOBr higher than that used here, whereas for 60%wt H2SO4 it reduces to 0.06. Similar differences can be found for other halogen species. This will alter the global chemical budget terms provided.

Despite efforts from experimentalists, there is a lack of experimental data on heterogeneous processing of halogens leading to large uncertainties and a need of

hypothesis and arbitrary choices to represent this heterogeneous chemistry (see e.g. Hossaini et al. 2016, Badia et al. 2019). We chose to build on the work of other modelers who gathered experimentally constrained uptake coefficients to build their model or justified hypothesis when data where missing. References for Table 5 are now added.

**Review #2.** The chemical mechanism mainly includes gas-phase chemistry, and thus lacks some heterogeneous chemistry that has been demonstrated to be important for reactive halogen production and loss. Nonetheless, this represents an important step forward for a global chemistry-climate model. Tropospheric reactive halogens have been shown to be an important lever for controlling the oxidation capacity of the atmosphere, so chemistry models should include this chemistry. The chemistry is complex and has large impacts on ozone, making it a somewhat scary endeavor. Thus, I commend the authors for taking this step forward in their model.

They mainly compare their model with other models that have a more complex representation of tropospheric reactive halogen chemistry, CAM-Chem and GEOS-Chem. Despite lacking important heterogeneous chemistry, their model agrees reasonably well with these other models, with some discrepancies that they attempt to explain. Results comparing preindustrial and industrial simulations and the impact of reactive halogen chemistry during these two time periods also is similar to these other models.

My main suggestions are to clarify the chemistry that is included in their model. Specific examples are listed below:

- In Figure 1, does "chlorine contribution to bromine recycling" indicate the three reactions above this statement, or other reactions? If the latter, which reactions are these? Are they in Table 5?

Yes the sentence refers to the 3 reactions (as for the box above, in which "Methane consumption" is the 'translation' of the Cl+VOC/CH4)

- Also in Figure 1, "heterogeneous reactions of bromine on sea salt and sulfate aerosols" is stated, but it would be helpful to refer to specific reactions in the tables. I don't see these reactions in Table 5.

This box refers to the reactions listed in Table 5. The figure 1 is a very simplistic representation of the halogenated chemistry and the main steps to represent in a global model, the aim here is to introduce the paper to non experts but all the details are in the Tables.

- Table 5 lists the reaction probability for heterogeneous reactions. The reaction probability formulation is typically used for first order reactions, but some reactions listed here are second order, so I'm confused. More detail is needed on how the reaction probability is used for these second-order reactions.

Second order reaction rate constants are calculated by assuming that the first reactant is limiting thus, the first-order rate constant is divided by the concentrations of the adsorbed species as commonly done in models (e.g. GEOS-CHEM and WRF-CHEM). We added this information in the paper.

- Sea salt chloride is emitted at HCl. Why is sea salt bromide not emitted as HBr? Sea-salt bromide is the largest primary source of bromine to the troposphere.

Seasalt bromide is emitted mainly as BrCl and $Br_2$, as represented in the model, the emission of HBr from seasalt is minimal, unlike that for HCl (see Saiz-Lopez and von Glasow, Chemical Society Reviews, 2012) In addition, the process is poorly understood, its quantification highly uncertain and some simulations have shown higher discrepancies, in particular in the representation of the vertical gradient of BrO in the tropic probably due to missing processes in the representation of the halogen chemistry (Schmidt et al. 2016). For that reason, this source is not systematically included in model studies (Sherwen et al. 2016, Badia et al. 2019)

Schmidt, J. A., et al. (2016), Modeling the observed tropospheric BrO background: Importance of multiphase chemistry and implications for ozone, OH, and mercury, *J. Geophys. Res. Atmos.*, 121, 11,819– 11,835, doi:10.1002/2015JD024229.

- Figure 5 says that reactive bromine is produced from N2O5, but I don't see this reaction in Table 5. It is mentioned in the text that N2O5 does not produce ClNO2, but this is inconsistent with Table 5.

Thank you for mentioning that. Actually, N2O5 had been written by mistake on Figure 5 but forgotten from figure 6. It is now fixed. The text was OK explicitly saying that "$ClNO_2$ is emitted by the heterogeneous reaction of $N_2O_5$ on sea salts and produces 2217 Gg $Cl.yr^{-1}$."

---

## Author Response (AR3)

The authors are thankful to Rafael Pedro Fernandez for his careful review of the paper.

All the minor comments were addressed through changes in the new version of the paper. Regarding the first main comment, the net inorganic bromine source from sea-salt is actually included (as can be seen in the reactions listed in Table 5). Regarding the inorganic iodine distribution (second main comment), they have been pre-computed by CAM-Chem, which includes the parameterization of Prados-Roman et al. (2015) allowing to account for ozone changes between preindustrial and present-day conditions. The description has been rephrased to make it clearer.